# Interaction dynamics between innate and adaptive immune cells responding to SARS-CoV-2 vaccination in non-human primates

Chaim A. Schramm [1,3], Damee Moon [1,3], Lowrey Peyton [1,3], Noemia S. Lima[1,3], Christian Wake[1,3], Kristin L. Boswell[1], Amy R. Henry[1], Farida Laboune[1], David Ambrozak[1], Samuel W. Darko[1], I-Ting Teng[1], Kathryn E. Foulds [1], Andrea Carfi [2], Darin K. Edwards [2], Peter D. Kwong [1], Richard A. Koup[1], Robert A. Seder [1] ✉ & Daniel C. Douek [1] ✉

As SARS-CoV-2 variants continue evolving, testing updated vaccines in non-human primates remains important for guiding human clinical practice. To date, such studies have focused on antibody titers and antigen-specific B and T cell frequencies. Here, we extend our understanding by integrating innate and adaptive immune responses to mRNA-1273 vaccination in rhesus macaques. We sorted innate immune cells from a pre-vaccine time point, as well as innate immune cells and antigen-specific peripheral B and T cells two weeks after each of two vaccine doses and used single-cell sequencing to assess the transcriptomes and adaptive immune receptors of each cell. We show that a subset of S-specific T cells expresses cytokines critical for activating innate responses, with a concomitant increase in CCR5-expressing intermediate monocytes and a shift of natural killer cells to a more cytotoxic phenotype. The second vaccine dose, administered 4 weeks after the first, elicits an increase in circulating germinal center-like B cells 2 weeks later, which are more clonally expanded and enriched for epitopes in the receptor binding domain. Both doses stimulate inflammatory response genes associated with elevated antibody production. Overall, we provide a comprehensive picture of bidirectional signaling between innate and adaptive components of the immune system and suggest potential mechanisms for the enhanced response to secondary exposure.

Severe acute respiratory syndrome coronavirus 2 (SARS-CoV-2) is the causative agent of coronavirus disease 2019 (COVID-19), which was declared a global pandemic in March 2020 by the World Health Organization[1]. Just 2 months after the original SARS-CoV-2 genomic sequence was published in January 2020, the Moderna mRNA-1273 vaccine candidate entered phase I clinical trials in humans[2], eventually proving 94% effective at preventing symptomatic infection with the ancestral SARS-CoV-2 virus[3]. Based on these and other data, mRNA-1273 is currently approved for use in at least 88 countries[4]. A similar mRNA vaccine from Pfizer, BNT162b2, is also in wide use.

Rhesus macaques (*Macaca mulatta*) serve as a model organism for the human immune system, and their similarity to humans as well

[1]Vaccine Research Center, National Institute of Allergy and Infectious Diseases, National Institutes of Health, Bethesda, MD 20892, USA. [2]Moderna Inc., Cambridge, MA 02139, USA. [3]These authors contributed equally: Chaim A. Schramm, Damee Moon, Lowrey Peyton, Noemia S. Lima, Christian Wake. ✉e-mail: rseder@mail.nih.gov; ddouek@mail.nih.gov

as outbreeding make them valuable for vaccine research[5]. Rhesus macaques were quickly identified as a good model of SARS-CoV-2-induced respiratory disease[6] and were critical in developing the original vaccines against SARS-CoV-2[7–10]. Macaques have also been used extensively in the testing of updated vaccines based on emerging variants[11,12].

The primary series of mRNA-1273 consists of two doses of 100 μg each, given four weeks apart. Much effort has been spent on characterizing the adaptive immune responses elicited by the vaccine. These studies have revealed that serum antibody titers remain detectable through at least 6 months after vaccination[13]. The frequency of S-binding memory B cells in the periphery increases between 3 and 6 months after vaccination[13], concurrent with an ongoing increase in somatic hypermutation[14]. T cell responses are also stable through at least 6 months[13,15], with the CD4 response polarized toward a Th1 phenotype[2,13]. T cells elicited by mRNA-1273 vaccination are also highly cross-reactive against later variants of concern, including Omicron[15,16].

Despite the wealth of data on adaptive immunity, innate immune responses to vaccination with an mRNA-based antigen and their interactions with adaptive immune responses remain less understood. Reports have differed on whether intermediate[17,18] or classical monocytes[19] increase in frequency in response to BNT162b2-vaccination, as well as on whether natural killer (NK) cells are mobilized or not[18,20]. BNT162b2 also appears to enhance the response of interferon-stimulated genes and the production of cytokines such as CXCL10 and interferon gamma (IFNγ)[17,19] in the 24 hours after vaccination. Similar data on the response to mRNA-1273 vaccination have not been available to date. A systems analysis of the innate and adaptive immune responses to BNT162b2 found that the second dose elicited a substantially heightened innate immune response than the first dose and revealed specific innate immune pathways that were linked to the quality of the adaptive immune response[17]. Another recent study also identified pre-vaccination innate immune states that correlated with higher antibody responses to other vaccines[21]. These comprehensive analyses offered key understandings of the overall immune response to vaccines.

Employing an integrated systems vaccinology approach can offer holistic novel insights to vaccine responses[22]. Here, we present an analysis of the immune response to the mRNA-1273 vaccine in non-human primates that integrates innate, T cell, and humoral arms. We find that both primary vaccine doses produce strong activation of all components of the immune system. This stimulation further increases after the second dose, with an emphasis on genetic pathways that link innate and adaptive immunity, such as those involving tumor necrosis factor alpha (TNF), IFNγ, interleukin 21 (IL-21), and C-C motif chemokine ligand 3 (CCL3). These insights allow mechanistic hypotheses for the success of mRNA-1273 that can be applied to the development of future vaccines.

## Results

### Cross-species convergence in B cell responses

Peripheral blood mononuclear cells (PBMCs) were collected from eight rhesus macaques from a preclinical trial of mRNA-1273 immunization[7] at baseline and 2 weeks after each of two doses of 100 μg mRNA-1273 vaccine, given 4 weeks apart (Fig. S1a). We sorted individual antigen-specific B cells from PBMCs to interrogate the dynamics of the humoral response (Fig. S1b). As expected, almost no antigen-specific cells were detected in baseline samples (Fig. 1a, b). IgM memory cells responding to the vaccine peaked after the first dose, while the frequency of antigen-specific IgG cells increased substantially after the second dose (Fig. 1a, b). Similar to previous reports in nonhuman primates (NHPs)[23], the response was primarily directed to epitopes on the Spike (S-2P) other than the ones detected by receptor binding domain (RBD), N-terminal domain (NTD), and S1 probes after the first dose of vaccine, with increasing proportion of cells binding to the RBD probe after the second dose (Fig. 1c).

Individual IgG⁺ memory B cells that bound to either S-2P or RBD were sorted into plates (one cell per well) and IG was sequenced. We sorted a total of 3507 antigen-specific B cells, from 2089 of which we recovered paired heavy and light chain sequences. This revealed highly polyclonal repertoires that did not overlap substantially between time points (Fig. 2a and Fig. S2). Unexpectedly, there was no significant difference in heavy chain somatic hypermutation (SHM) levels

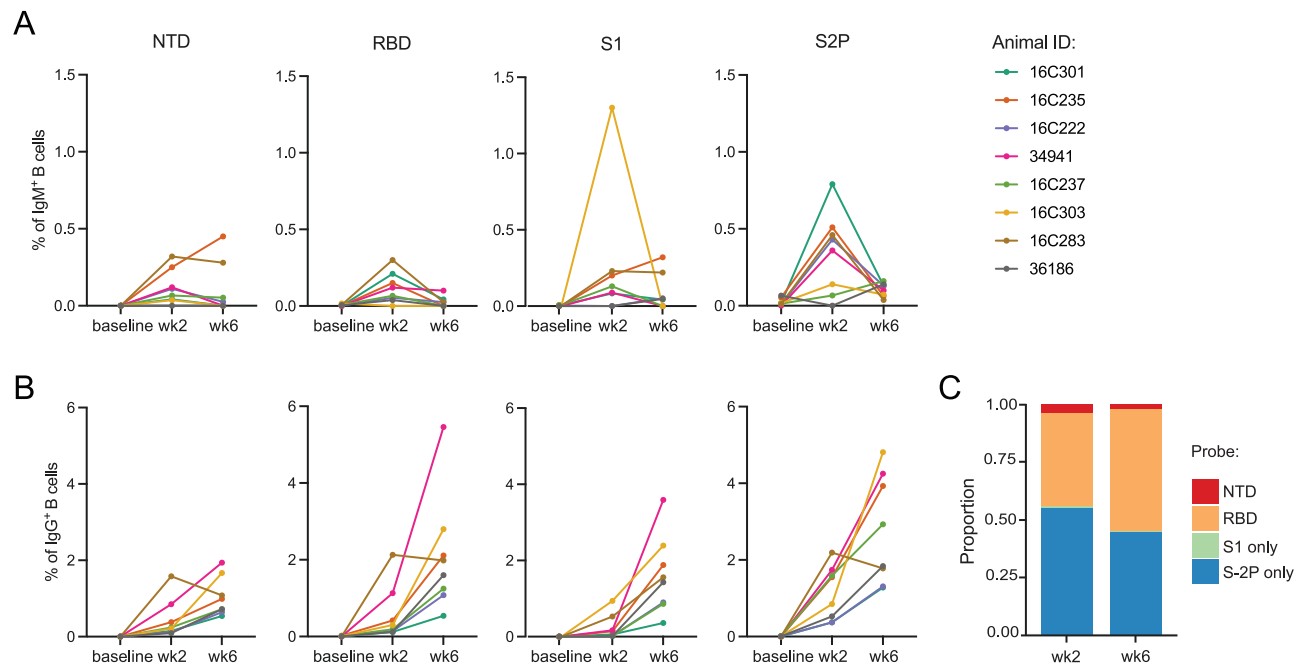

**Fig. 1 | Frequency dynamics of antigen-specific B cells on vaccinated animals.** **A** Frequencies of memory IgM (IgD⁻IgM⁺) B cells and **B** IgG B cells binding to SARS-CoV-2 probes (NTD, RBD, S1 or S2P) from eight animals. **C** Proportion of antigen-specific B cells binding to each probe (cells from eight animals combined). Source data are provided as a Source Data file.

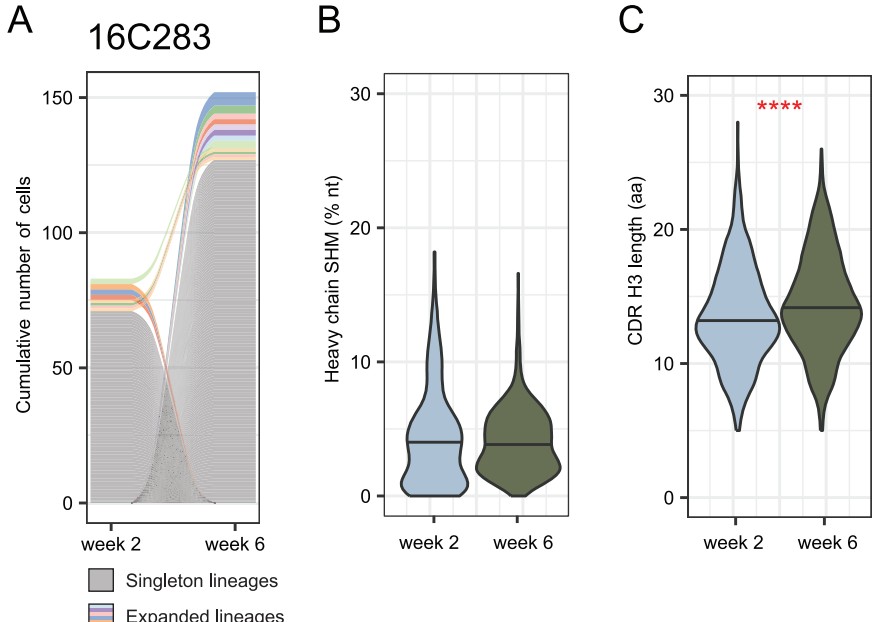

**Fig. 2 | Antigen-specific B cell repertoire of vaccinated animals. A** Representative alluvial plot from B cell clonal dynamics from one animal. Gray lines represent singleton B cells and colored lines represent expanded lineages. Lines that start or end at zero, including all singletons, indicate B cell lineages that were observed in only a single time point. Colored lines that span the graph represent lineages found at both time points. The thickness of each line at either side of the panel is proportional to the number of cells in that lineage at the corresponding time point, with the cumulative number of cells in all lineages indicated on the y-axis. $N = 83$ cells from week 2 and 152 cells from week 6. Similar plots for other animals may be found in Fig. S2. **B** Heavy chain somatic hypermutation (SHM) of all antigen-specific memory B cells from eight animals. $N = 696$ cells at week 2 and 1393 cells at week 6. Distributions were compared using 2-sided unpaired Wilcoxon test. Distributions for individual animals may be found in Fig. S3a. Distributions considering only one cell per lineage may be found in Figs S3b and S3c. **C** CDR H3 length in amino acids (aa) of antigen-specific memory B cells from 8 animals. Because CDR H3 length is essentially invariant within a lineage, only one cell per lineage was included. $N = 641$ lineages at week 2 and 1238 lineages at week 6. Distributions were compared using a 2-sided unpaired Wilcoxon test, $P = 1.2E-7$. Distributions for individual animals may be found in Fig. S4a. Distributions considering all cells may be found in Figs. S4b and S4c. Source data are provided as a Source Data file.

between time points (Fig. 2b and Fig. S3a); however, we did observe slight increases in light chain SHM at week 6 (Fig. S5). We also found longer CDR H3 loops after the second vaccination compared to the first (Fig. 2c and Fig. S4).

Similar to observations in humans, we found differential usage of several V genes in the antigen-specific memory B cells compared to unselected naïve B cell repertoires from the baseline samples (Fig. 3a and Fig. S6). Three $V_H$ genes were significantly enriched in the antigen-specific repertoire: *IGHV1-AAU*, *IGHV4-ADG*, and *IGHV4-AGR*, as well as $V_K$ genes *IGKV2-AAO* and *IGKV4-ACX* and $V_L$ genes *IGLV1-ABN*, *IGLV3-AAV*, *IGLV3-AEC*, and *IGLV3-AED*. Notably, the closest human homolog of *IGHV1-AAU* is *IGHV1-69* (Fig. S7), which has been reported as a major public clone binding to the S2 domain on Spike[24,25]. Thus, while the immunogenetic diversity of macaques is much greater than in humans[26,27], they nevertheless produce stereotyped antibody responses to SARS-CoV-2 that parallel those observed in humans.

We next identified public clones, defined here as *IgH* sequences from different animals using the same $V_H$ gene and having at least 80% amino acid identity in CDR H3. While there is no single consensus definition of a B cell public clone, this threshold has been used previously and is thought to permissively capture antibodies with functionally similar antigen binding profile[24,25,28]. We found two such public clones that were present in at least four of the eight animals, both using *IGHV3-AFR* (Fig. 3b, c). Although these comprise only 14 of over 2000 cells analyzed and are thus not a dominant component of the antibody response, they still represent an unexpected convergence. By comparison, out of 30,556 naïve B cells sorted from these 7 of these 8 animals, only two public clones comprising just 11 cells were identified using similar criteria.

Notably, *IGHV3-AFR* is the closest macaque homolog of human *IGHV3-30* and closely related genes such as *IGHV3-30-3*, *IGVH3-30-5*,

and *IGHV3-33* (Fig. S7), at least the first two of which are enriched in the anti-SARS-CoV-2 response[29,30]. The first public clone has a relatively short CDR H3 of nine residues, enriched for small amino acids (Fig. 3b). Strikingly, the second public clone matches the signature of an *IGHV3-30*-derived public clone that has been described in humans[24,25] (Fig. 3c). Members of this public clone have been shown to bind an epitope in the S2 domain, consistent with our data in which these clones were found to bind exclusively to S-2P probe and not to probes corresponding to S1 domain (S1, RBD, and NTD probes). These remarkable overlaps with the human antibody response show that mRNA-1273 vaccination elicits a consistent, reproducible adaptive immune response.

## Increase in circulating germinal center-like B cells after second vaccine dose

We sequenced the transcriptomes of the sorted antigen-specific B cells to examine the phenotypes of the cells elicited by immunization. After quality control, we analyzed 2034 cells, including 1757 for which we had previously recovered heavy and light chain IG sequences. UMAP clustering revealed four populations of cells (Fig. 4a), the largest of which consists of highly activated cells with a light zone (LZ)-like phenotype, including the expression of *CCL3*, *BCL2A1*, and *MYC* (Fig. 4b). There were two clusters of activated memory (AM) B cells, both expressing similar levels of markers like *TOX2*, *FGR*, and *ITGAX* (Fig. 4b). Closer inspection of these two clusters showed that they differed primarily in the expression of unannotated light chain V genes or pseudogenes, such as *ENSMMUG00000047452* and *ENSMMUG00000055402*. As differences in V gene usage are not linked to transcriptomic phenotypes[31], we combined these two clusters into a single AM cluster for downstream analysis. The final cluster was enriched for the expression of *CR2* (CD21), *LTB*, *TCF7*, and other genes which have

A

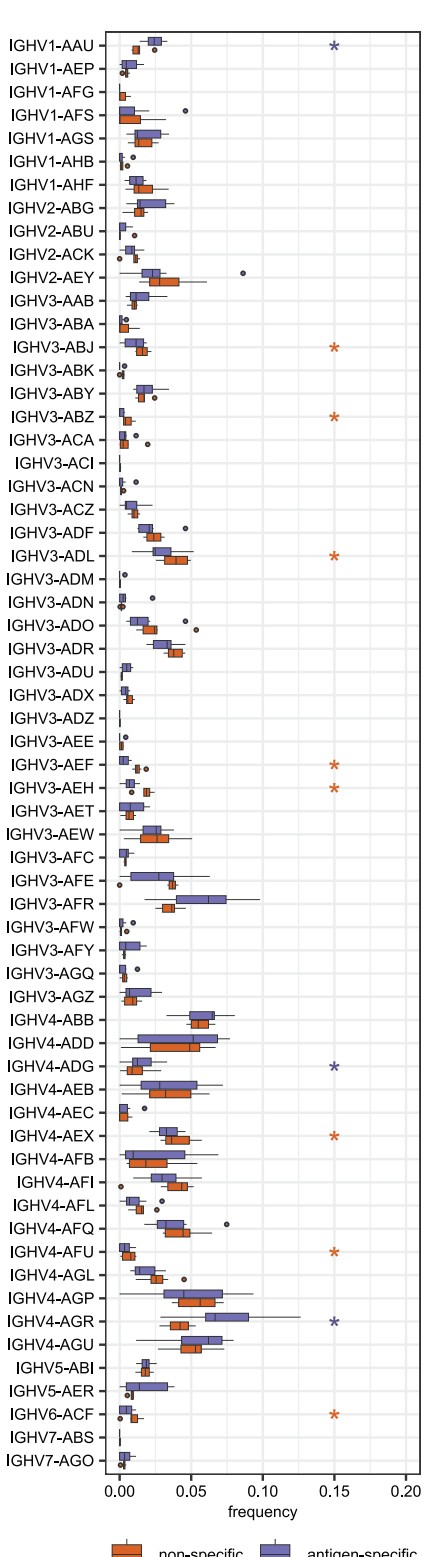

non-specific    antigen-specific

B

| animal | time | specificity | V gene | CDRH3 |
|--------|------|-------------|--------|-------|
| 16C222 | wk2 | S-2P | IGHV3-AFR | CARGSGSSDYW |
| 16C222 | wk6 | S-2P | IGHV3-AFR | CARGSGSSDYW |
| 16C222 | wk6 | S-2P | IGHV3-AFR | CARGSGS**A**DYW |
| 16C237 | wk2 | S-2P | IGHV3-AFR | CARGSG**LG**DYW |
| 16C283 | wk6 | S-2P | IGHV3-AFR | CARGSGSSD**R**W |
| 34941 | wk2 | S-2P | IGHV3-AFR | CARGSGSSDYW |
| 34941 | wk2 | S-2P | IGHV3-AFR | CARGSGSSDYW |
| 36186 | wk2 | S-2P | IGHV3-AFR | CARGSGSGDYW |
| 36186 | wk6 | S-2P | IGHV3-AFR | CARGSGP**P**GDYW |

C

| animal | time | specificity | V gene | CDRH3 |
|--------|------|-------------|--------|-------|
| 16C235 | wk6 | S-2P | IGHV3-AFR | CARA**H**GGSYYYG**F**D**C**W |
| 16C237 | wk2 | S-2P | IGHV3-AFR | CAR**P**RGGSYYYGLDSW |
| 16C283 | wk6 | S-2P | IGHV3-AFR | CARA**H**GG**T**YYYGLDSW |
| 16C301 | wk6 | S-2P | IGHV3-AFR | CAR**G**RGG**N**YYYGLDSW |
| 16C301 | wk6 | S-2P | IGHV3-AFR | CARARGGSYYYGLDSW |
| *Human* | | | *IGHV3-30* | AR..GSY..FD. |

previously been identified as being upregulated in resting memory (RM) B cells compared to AM[32].

Relative to week 2, the week 6 time point had more LZ-like cells and fewer RM (Fig. 4c), again demonstrating the stronger immune activation achieved by the second vaccine dose. Serum neutralization increases substantially after the second dose[2,7], which is largely associated with antibodies targeting epitopes in RBD and NTD[33]. We

observed that LZ-like cells are enriched for RBD- and NTD-binding B cells (Fig. 4d); together with the temporal dynamics of these clusters, this is consistent with the identification of these cells as having recently exited from germinal centers (GCs) in response to the second dose of vaccine. Despite this, we did not find any differences in SHM levels among the B cell clusters (Fig. 4e), perhaps due to the lack of pre-existing immunity and the short duration of affinity maturation. In

**Fig. 3 | V gene usage and public clones. A** Comparison of VH gene usage between all naïve B cells (orange) and antigen-specific memory B cells (purple) from seven animals (animal 16C237 was excluded due to lack of naive repertoire data). Boxes show the interquartile range, with the median marked as heavy horizontal band. Whiskers represent the highest (lowest) datapoint within 1.5 times the interquartile range of the 75th (25th) percentile. *N* = 1706 antigen-specific lineages and 29,161 sorted naive B cells. Genes with statistically significant differences by a 2-sided paired Wilcoxon test are indicated with asterisks in the color of the up-regulated group. *P* = 0.016 for *IGHV1-AAU*, *IGHV3-AEF*, *IGHV4-AFU*, *IGHV4-AGR*, and *IGHV5-AER*. *P* = 0.031 for *IGHV3-ABZ*, *IGHV3-AEH*, and *IGHV4-ADG*. *P* = 0.047 for *IGHV3-ABJ*, *IGHV3-ADL*, and *IGHV4-AEX*. **B** A public B cell clone using *IGHV3-AFR* that was found in five of eight vaccinated animals. Residues that differ from the consensus are highlighted in red. Two identical rows indicate two cells with the same amino acid CDR H3 found in a single animal and time point. **C** A second public clone was found in four animals. The consensus of a corresponding human public clone is shown as a logo plot. Conserved positions in the human public clone, which are hypothesized to be functionally important, are also conserved in the rhesus public clone. Source data are provided as a Source Data file.

addition, LZ-like cells were somewhat more clonally expanded than AM and RM cells (Fig. 4f), though all three clusters are dominated by cells from unexpanded lineages.

We further analyzed longitudinal transcriptomic changes within each cell type using Gene Set Enrichment Analysis (GSEA)[34], conducted with the Hallmark gene set collection from the Molecular Signatures Database (MSigDB)[35]. This revealed eleven pathways with significant association to the cluster-stratified time point differential expression analyses (Fig. 4g). Nine of these were uniquely elevated in LZ-like B cells, including activation-related pathways such as IFN Gamma Response, IL-2 STAT 5 Signaling, and Inflammatory Response. LZ-like B cells also showed increased expression of the Hypoxia pathway after the second dose of vaccine, aligning with recent GC exit. The GC, and the light zone in particular, is known to be a hypoxic environment, which has been shown to promote class-switch recombination and drive differentiation toward a CD27+ memory phenotype[36,37]. This suggests that the second vaccine dose serves to amplify the pool of S-specific memory B cells available for a protective anamnestic response in the future[38].

Notably, the TNF Signaling via NFκB pathway was significantly upregulated across all B cell subtypes at week 6 (Fig. 4g), though *TNF* itself was highly expressed only in LZ-like B cells (Fig. S8a). TNF production by B cells is stimulated by both Ig- and CD40-induced activation and is critical for B cell proliferation[39]. TNF and other proinflammatory cytokines secreted by B cells are also thought to enhance the response of innate effector cells during acute viral infection[40]. Thus, in stimulating the development of highly activated LZ-like B cells, the second dose of mRNA-1273 vaccination elicits cytokine responses that, together with other potential factors such as epigenetics and trained immunity, may initiate strong cooperation between humoral immunity and other immune cell types.

**Vaccine-elicited CD4 T cells influence both humoral and innate immune response**

It was previously reported that the primary series of mRNA-1273 induces a robust CD4 response that is strongly polarized toward Th1 and Tfh phenotypes, with somewhat lower CD8 responses[41,42]. To characterize the TCR repertoire and transcriptome profile of CD4 T cells elicited by mRNA-1273, PBMCs from weeks 2 and 6 were stimulated with SARS-CoV-2 S1 and S2 peptide pools comprising both domains of the S protein[7]. S-specific memory CD4 T cells were sorted based on upregulation of the activation markers CD154 (CD40L) and CD69 (Fig. S1c). As a control, non-specific memory CD4 T cells were also sorted from the non-stimulated (DMSO only) condition (Fig. S1c). Frequencies of S-specific CD4 T cells increased in all but one animal at week 6 compared to week 2, by an average of 3-fold for both S1 and S2 stimulations (Fig. 5a), as expected for a secondary response.

A total of ~8000 S-specific CD4 T cells and 24,000 non-specific cells were loaded into a 10× Chromium controller and the transcriptomes and T cell receptors (TRs) were sequenced. After pre-processing and quality control, we recovered transcriptomes and TRB sequences from 664 S-specific and 1259 non-specific cells; TRA sequence recovery was too low to analyze further. We did not observe any significant clonal expansion in the S-specific TRB. This may partially be due to low TR recovery but is also consistent with previous reports that BNT162b2 vaccination produces relatively little clonal expansion in T cells[43]. A few TRBV genes were used at different rates in S-specific vs non-specific T cells (Fig. 5b), though only *TRBV3-4* was significantly enriched in S-specific cells. *TRBV3-4* is most closely related human *TRBV3-1*; notably the frequency of TRBV3 usage has been observed to be enriched in repertoires from cohorts of COVID-19 patients compared to healthy controls[44].

We then examined the transcriptomes of CD4 T cells, sorted as S-specific (CD154+CD69+ upon stimulation with spike peptides) or non-specific (CD154−CD69− from no peptide control). After integration of all sorted CD4 T cells using Harmony[45], UMAP analysis defined four clusters (Fig. 5c), which aligned well with the experimental setup (Fig. 5d). Two clusters had an activated phenotype, based on canonical markers, and two had a resting profile (Fig. 5e). One small cluster of resting cells was characterized by expression of genes related to cellular stress (*DNAJB1* and *UBB*), most likely due to manipulation of cells during the experimental processing. The larger cluster of resting cells represents the total population of memory CD4 T cells that are not S-specific. Due to limited mRNA recovery (median of 753 UMIs per cell), we were unable to identify specific T-cell memory subsets.

Both clusters of stimulated T cells showed increased expression of the genes *BCL2A1*, *FABP5*, *CD40LG* and *IL21*. *BCL2A1* and *FABP5* encode proteins (Bcl-2 and Fatty Acid Binding Protein 5) that are involved in long-term survival of memory T cells and confirm the differentiated state of these antigen-specific CD4 T cells. CD40L (or CD154) was one of the markers we used for sorting antigen-specific T cells because it is upregulated in activated CD4 T cells and has an important role in providing signals to B cells during GC development. Interestingly, *IL21* was expressed in both clusters of stimulated T cells with higher expression in the smaller cluster. This cytokine is produced by several CD4 T cell subsets, including Tfh, Th17, and Th9, and has a broad immunological effect that can activate various cell types such as NK cells, monocytes, dendritic cells, CD4 and CD8 T cells, as well as B cells[46]. Importantly, it has a key role in Tfh differentiation, GC formation and maintenance, B cell differentiation into plasma cells, and class switching[47], which might be beneficial to establish a potent antibody response after vaccination.

In addition to *IL21*, the smaller cluster of stimulated CD4 T cells exhibited increased expression of genes related to production of the chemokines CCL3 and CCL4 and cytokines IL-10 and IFNγ. CCL3 and CCL4, also known as Macrophage Inflammatory Protein (MIP)-1α and MIP-1β, respectively, can act as chemoattractants for monocytes/macrophages, NK cells, dendritic cells, and T cells from the circulation to inflamed tissue[48]. Co-expression of CCL3 and CCL4 has been demonstrated to guide naïve CD8 T cells to sites of CD4 T cell and antigen-presenting dendritic cell interaction[49]. It has been demonstrated that IL-10 blockade in NHPs enhances inflammation in the lungs caused by SARS-CoV-2 infection and, interestingly, the lack of IL-10 impairs accumulation of tissue resident memory T cells[50]. Thus, the upregulation of this anti-inflammatory cytokine by S-specific CD4 T cells may be contributing to the balanced immune response induced by the vaccine by preventing exacerbated inflammation,

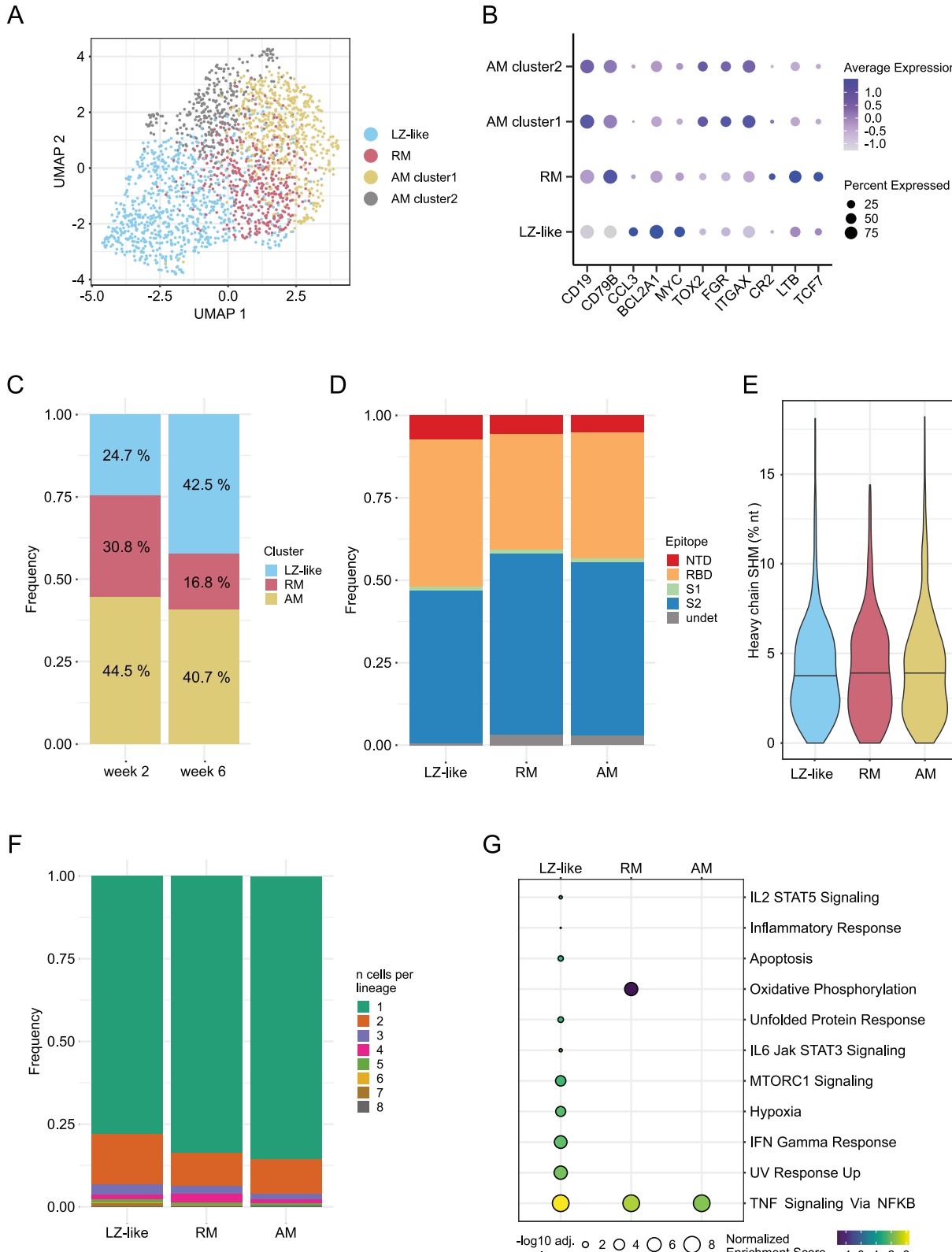

while IFNγ promotes activation of innate and adaptive immune responses. Indeed, IFNγ expression has been widely described in response to SARS-CoV-2 infection and vaccination[51]. These results demonstrate that vaccine-induced CD4 T cells contribute to overall immune activation triggered by mRNA-1273 vaccination and may suggest a late cross-talk between the adaptive and innate immune responses.

## Phenotypic differences in innate immune cells elicited by the second vaccine dose

Innate immune responses have traditionally been investigated in the first hours after vaccination and usually peak within 1-2 days. However, some studies have nevertheless described qualitative changes in innate immune subsets in response to a booster vaccination compared to the initial dose[52,53]. Here, we examined the phenotypes

**Fig. 4 | B cell transcriptomics reveal a subset of highly activated cells that recently exited germinal centers. A** UMAP plot showing clusters based on gene expression from antigen-specific B cells. The clusters correspond to Light zone (LZ)-like cells, resting memory (RM) cells, and two activated memory (AM) clusters as shown in (**B**). **B** Differentially expressed genes that define the phenotype of each cluster identified in (A). Both AM clusters have identical functional markers and were combined for downstream analysis. **C** Frequencies of each cluster identified in (A) at each time point. **D** Epitope targeting as delineated by flow cytometry for each phenotypic cluster. **E** Heavy chain somatic hypermutation (SHM) for each cluster; no significant differences were found using a one-way ANOVA test. **F** Clonal expansion frequency for each cluster. **G** Gene set enrichment analysis comparing week 6 to week 2 for each cluster, using the Hallmark gene sets from MSigDB (34). A Kolmogorov-Smrinov test with the Benjamini–Hochberg correction for multiple testing was used to identify pathways with significant changes. Only pathways with at least one significant result are shown. A total of 2034 antigen-specific B cells sorted from 8 animals at weeks 2 and 6 were analyzed. Source data are provided as a Source Data file.

of innate cellular responses at the same time points as the adaptive immune response, 2 weeks after each vaccination. We sorted a total of 10,000 monocytes, natural killer (NK), and dendritic cells (DCs) from PBMCs from each sample (Fig. S1b) and generated transcriptomes using the 10x Chromium controller. After pre-processing and quality control, a total of 61,051 cells were recovered: 30,671 from the baseline time point, 23,073 from week two, and 7307 from week six. We used DeepImpute[54] to correct for low UMI counts and then integrated to remove technical and batch effects. After excluding doublets and low-quality cells, Louvain clustering detected ten clusters (Fig. 6a), which we annotated using canonical markers and differentially expressed genes (Fig. 6b). These included three monocyte clusters, two DC clusters, four NK clusters, and one platelet cluster.

Among the monocyte clusters, we identified classical, non-classical, and intermediate monocyte subtypes. Due to the panel of fluorochromes used, we were unable to quantify the frequencies of monocyte and NK cells among total PBMC from the flow cytometry data (Fig. S1b). Based instead on the transcriptomic data, the proportion of classical monocytes among all monocytes did not change significantly after vaccination, though the dynamics varied across animals (Fig. 7a). The proportion of non-classical monocytes showed a significant decrease after the second dose, compared to baseline, while intermediate monocytes progressively increased in proportion after each dose of the mRNA vaccine. The immune response to BNT162b2 in a study in humans demonstrated a similar pattern, with the frequency of intermediate monocytes peaking two days after each dose[17]. Intermediate monocytes are known to express high levels of CCR5[55], a receptor for CCL3 and CCL4 chemokines found in our study to be highly expressed by stimulated S-specific CD4 T cells and LZ-like S-specific memory B cells. This monocyte subset usually expresses the highest level of antigen-presentation related molecules and are efficient at producing pro-inflammatory cytokines[56]. Thus, the increased frequency of intermediate monocytes 2 weeks after vaccination may be due to chemokines produced by activated CD4 T cells, which might reciprocally help further activate T cells due to their high antigen-presentation capacity.

The four NK cell clusters were mainly comprised of two clusters that we labeled NK-1 and NK-2. Based on the transcriptional profile, both NK-1 and NK-2 clusters most likely represent an NK cell subset similar to CD56$^{dim}$ NK cell subset in humans, but at different levels of maturity. Increased expression of *EEF1A1*, *TPT1*, and *GZMB* defined the NK-1 cluster. NK-2, however, was characterized by *CXCL3* as well as complement factor genes *CFD* (Complement Factor D) and *C1QB* and higher expression of *FCGR3* (CD16) (Fig. 6c). The higher expression of CD16, an effector molecule related to NK cell cytotoxicity, suggests that NK-2 represents a more mature state of the NK-1 cells. While NK cells were polarized toward the NK-1 phenotype at baseline, NK-1 and NK-2 frequencies were approximately equal after the first dose of vaccine, and half of the animals showed a dominance of the NK-2 phenotype after the second dose (Fig. 7b). Based on the more mature transcriptional profile of the NK-2 cells and the strikingly reciprocal dynamics of these two clusters, these data suggest that the NK-1 cells may be differentiating into NK-2 cells after the mRNA vaccination. The smallest NK cluster was identified as cycling NK cells because it showed

high expression of genes associated with cell proliferation (e.g., *STMN1* and *PCNA*). These three NK cell clusters all showed upregulation of the chemokine *CCL5* (Fig. 6b), which together with *XCL1* are critical for the recruitment of cDC1 cells[57].

The gene expression profile of the fourth cluster is consistent with that described for human CD56$^{bright}$CD16$^-$ (immature) NK cells[58], with high expression of *IL7R* and *XCL1* and low to no expression of *FCGR3* (CD16). Although we did not see upregulation of CD56 in these cells, CD56 is not a reliable marker of NK cell phenotype in NHPs[59]. CD56$^{bright}$ NK cells usually represent about 10% of the NK population in human peripheral blood and are thought to be precursors of the CD56$^{dim}$ NK cells[60]. CD56$^{bright}$ NK cells are known to be poorly cytotoxic, but easily activated due to higher expression of cytokine receptors and are very efficient at proliferating and producing cytokines, particularly IFNγ. Upon stimulation with cytokines, they preferentially migrate to secondary lymphoid organs whereas CD56$^{dim}$ NK cells migrate to acute inflammatory sites[60]. This cluster decreased in frequency after the initial prime but were highly variable after the second dose (Fig. 7b), suggesting a vaccine-induced differentiation of NK cells from this immature state to a more mature phenotype. This aligns with previous reports finding that NK cell responses after a boost immunization with modified vaccinia virus Ankara (MVA) vaccine showed a more cytotoxic and mature phenotype than responses to the prime immunization[53].

The clustering also revealed plasmacytoid (pDC) and conventional DCs (cDCs), identified as expressing *IRF8* and either *IRF4* or *CLEC9A*, respectively (Fig. 6b). Unlike other innate cell types, the dendritic cell markers in our flow cytometry panel allowed for direct quantification of cDC and pDC frequencies among all PBMCs (Fig. S1b). We found a significant increase 2 weeks after the first dose for both cDC and pDC subsets (Fig. 7c). While cDCs showed a trend towards returning to baseline frequencies at the week 6 time point, pDC frequencies continued to be significantly higher than baseline (Fig. 7c). The ratio of conventional to plasmacytoid DCs did not change significantly between the time points tested (Fig. 7d). In contrast, pDCs in patients infected with SARS-CoV-2 decreased over time due to increased apoptosis, which correlated with disease severity[61], and an increased ratio of cDC to pDC has been reported in severe COVID patients[62]. Taken together, our data show that the mRNA-1273 vaccine induces changes in frequencies of several innate cell types that persist even 2 weeks after vaccination. We also detected high expression of genes induced by cytokines and chemokines produced by CD4 T cells at these later time points. We hypothesize that these CD4 T cell products are driving differentiation and migration of the innate cells, resulting in frequency changes. This suggests that the role of the innate cells in the immune response can persist for several days or weeks after immunization.

## mRNA-1273 activates an innate genetic program associated with high antibody responses

To further examine the pathways differentially expressed in response to the vaccine, we employed GSEA with the human Hallmark pathways from MSigDB. This analysis revealed that multiple inflammatory pathways were upregulated in week 6 compared to week 2 in most cell types except classical monocytes and immature NK cells (Fig. 8a).

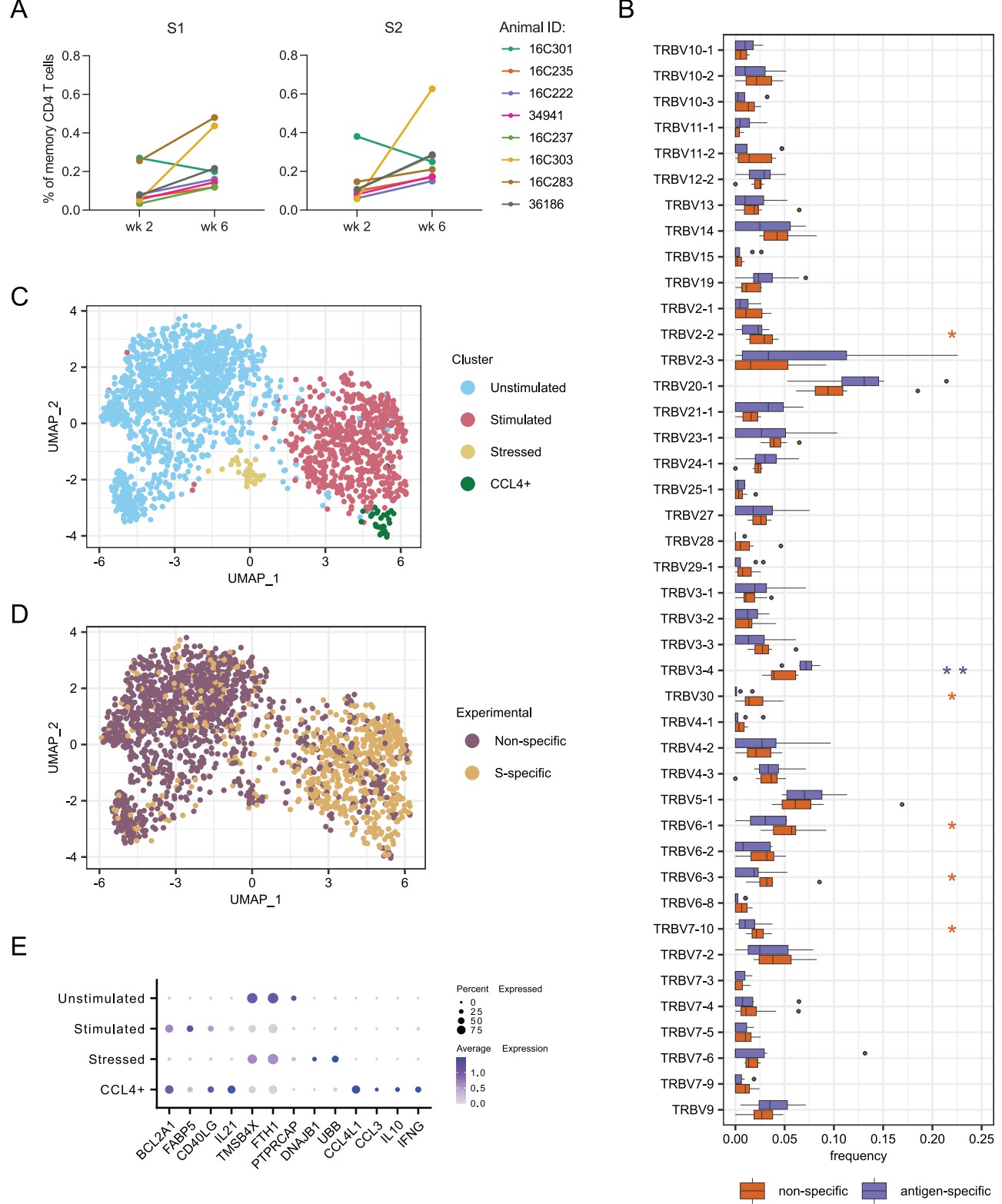

**Fig. 5 | T cell responses induced by vaccination. A** Frequencies of activated memory CD4 T cells after stimulation with peptide pools from S1 and S2 domains of SARS-CoV-2 spike. **B** Comparison of TRBV gene usage between non-specific and S-specific memory CD4 T cells sorted. Boxes show the interquartile range, with the median marked as heavy horizontal band. Whisker represent the highest (lowest) datapoint within 1.5 times the interquartile range of the 75th (25th) percentile. Genes with statistically significant difference by a 2-sided paired Wilcoxon test are indicated with asterisks in the color of the up-regulated group. *P* = 0.0391, 0.0078, 0.0225, 0.0391, 0.0391, and 0.0391 for *TRBV2-2*, *TRBV3-4*, *TRBV30*, *TRBV6-1*, *TRBV6-3*, and *TRBV7-10*, respectively. **C** UMAP plot identifying four phenotypic clusters based on gene expression. **D** UMAP plot colored by experimental conditions identifying S-specific and non-specific T cell by sort. These correspond closely to the phenotypic clusters defined in (**C**). **E** Differentially expressed genes that define the phenotype of each cluster identified in (**C**). A total of 664 S-specific and 1259 non-specific T cells sorted from 8 animals at weeks 2 and 6 were analyzed for all panels. Source data are provided as a Source Data file.

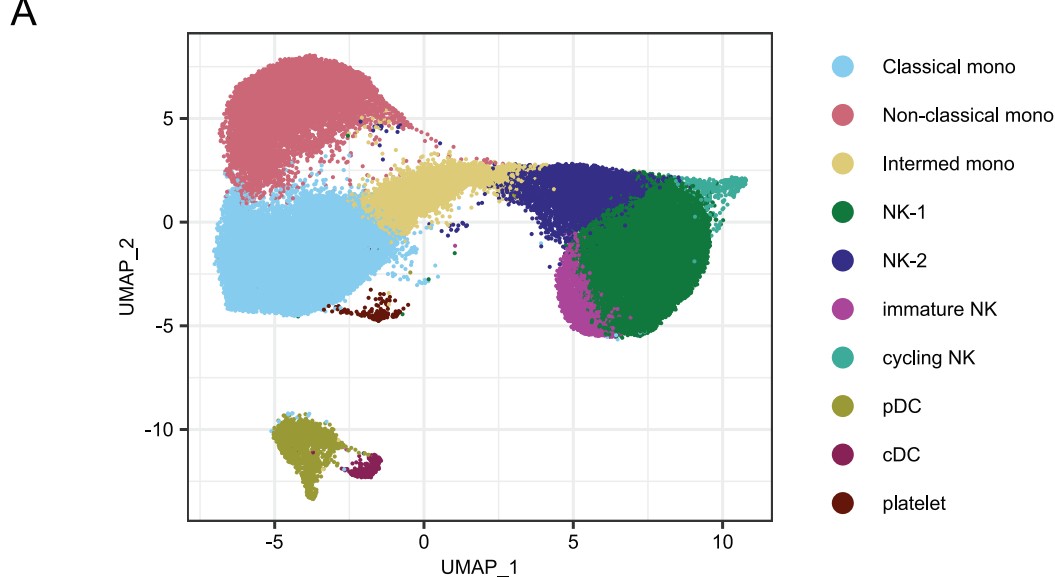

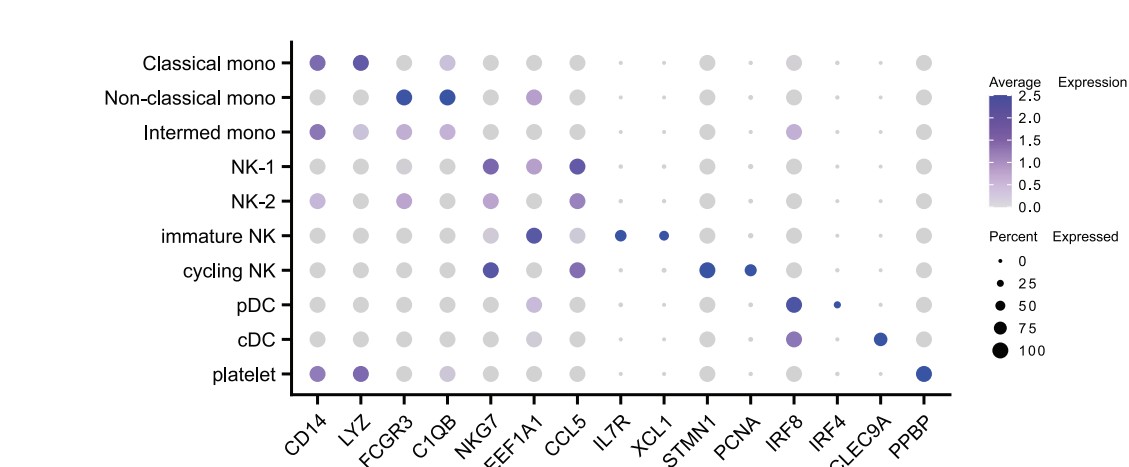

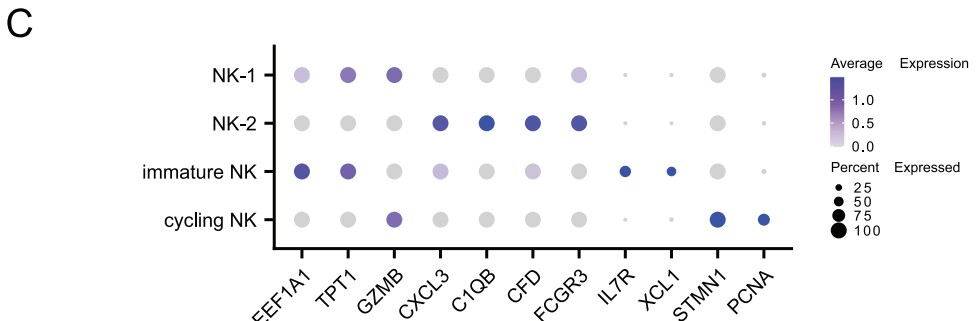

**Fig. 6 | Identification of innate immune cell subtypes. A** UMAP plot showing innate cell types based on clustering by gene expression. Mono monocytes, NK natural killer, pDC plasmacytoid dendritic cell, cDC classical dendritic cells. **B** Differentially expressed genes which define the clusters identified in (**A**). **C** Differentially expressed genes that define specific NK cell subsets. A total of 61,051 cells were analyzed.

This prominently includes TNF signaling via NF-κB (Fig. 8a), which was upregulated in all B cell subsets at week 6 as well (Fig. 4g). Similar to the pathways activated in LZ-like B cells, Inflammatory Response, IL6-Jak-Stat3 Signaling and, to a lesser extent, Interferon Gamma Response and Hypoxia were activated in multiple innate immune types (Fig. 8a). Notably, both TNF and IFNγ are involved in stimulating the presentation of antigen by monocytes[63,64], serving as bridges between the innate and adaptive immune responses.

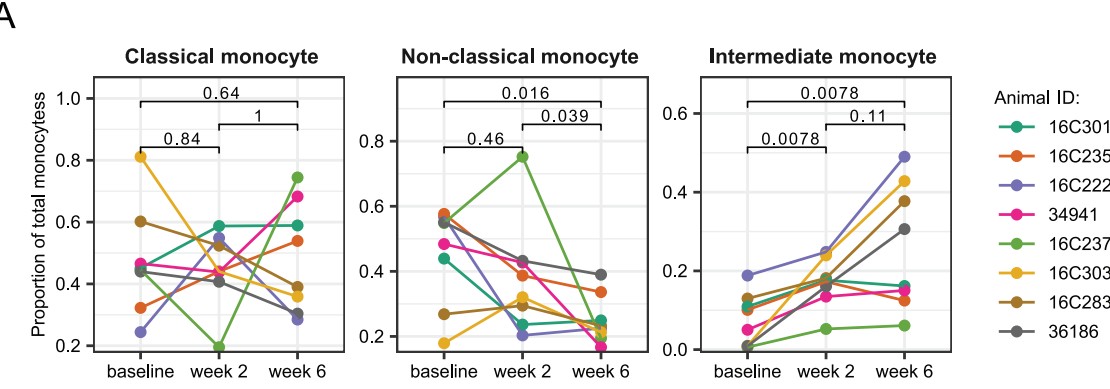

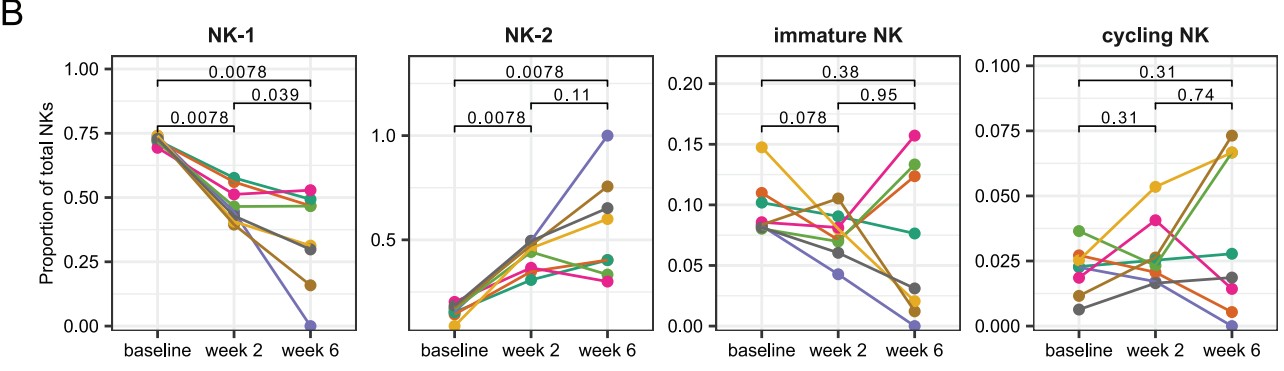

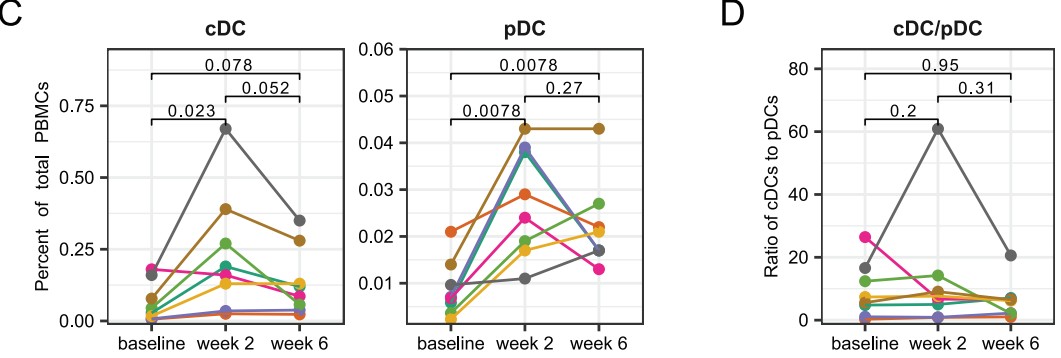

**Fig. 7 | Frequencies of cell types identified by gene signatures. A** Frequencies of monocyte subsets among all monocytes from sequencing data. **B** Frequencies of natural killer (NK) cell subsets among all NK cells from sequencing data. **C** Frequencies of dendritic cell (DC) subsets in total PBMC from flow cytometry data. **D** Ratio of classical DC (cDC) frequency to plasmacytoid DC (pDC) frequency.

A total of 30,671 cells from the baseline, 23,073 cells from week 2, and 7307 cells from week 6 were analyzed. P values were obtained using a 2-sided paired Wilcoxon test with Holm correction for multiple testing. Source data are provided as a Source Data file.

Increased IFNγ was also detected after the second vaccine dose in the plasma of people vaccinated with BNT162b2[17]. By contrast, SARS-CoV-2-infected people with severe disease had upregulation of the interferon beta pathway[65].

A recent study found that pre-vaccine enrichment of several of these pathways was associated with higher antibody production across a range of vaccine types[21]. We used the 50 inflammatory marker genes that were identified as distinguishing between high and low vaccine responses to generate a module score in Seurat (Table S1). Analysis of our data revealed that the vaccine-predictive inflammatory response increased over time in many cell types apart from classical and intermediate monocytes (Fig. 8b). In the original study, expression of the gene module was associated with the frequencies of classical monocytes; here we observed the absolute scores were highest for classical monocytes at the baseline time point, perhaps consistent with the pre-vaccine time point being most important for predicting antibody responses. However, as all of the animals in this study produced high serum Ig responses over a relatively narrow range[7], we were unable to validate the correlation of this signature with that outcome. Nonetheless, it is striking that the temporal dynamics of the expression of this vaccine-predictive inflammatory module in most innate cell types match those observed for the adaptive immune response. This suggests that residual activation from the first dose, as reflected in expression of these markers, is a critical factor in the increased B cell responses elicited by the second dose of mRNA-1273.

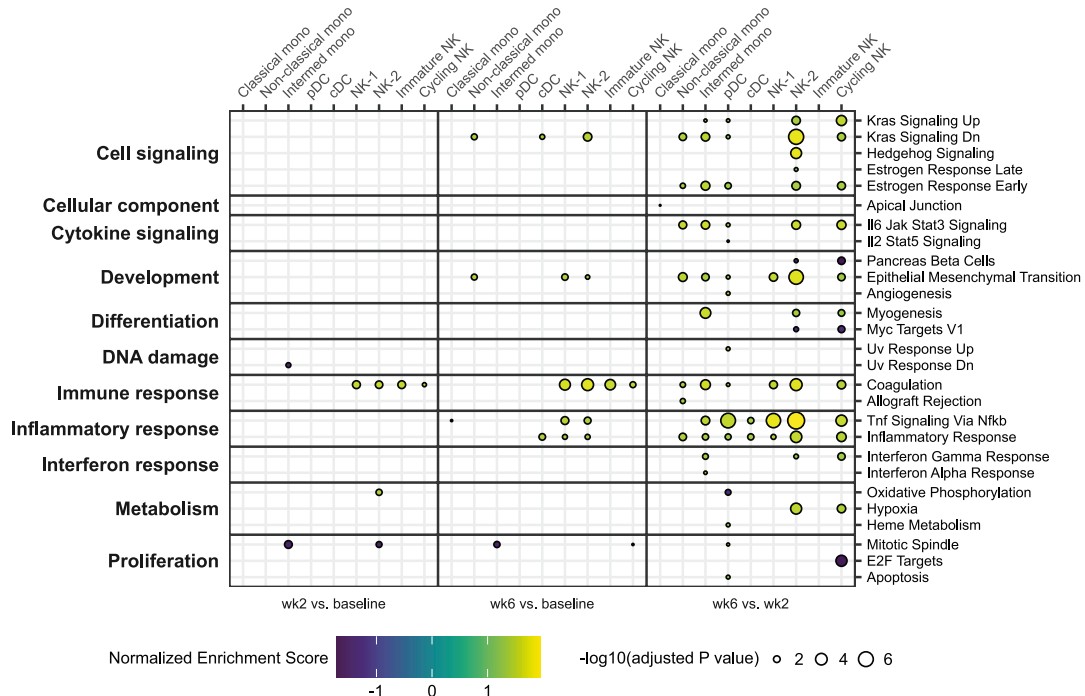

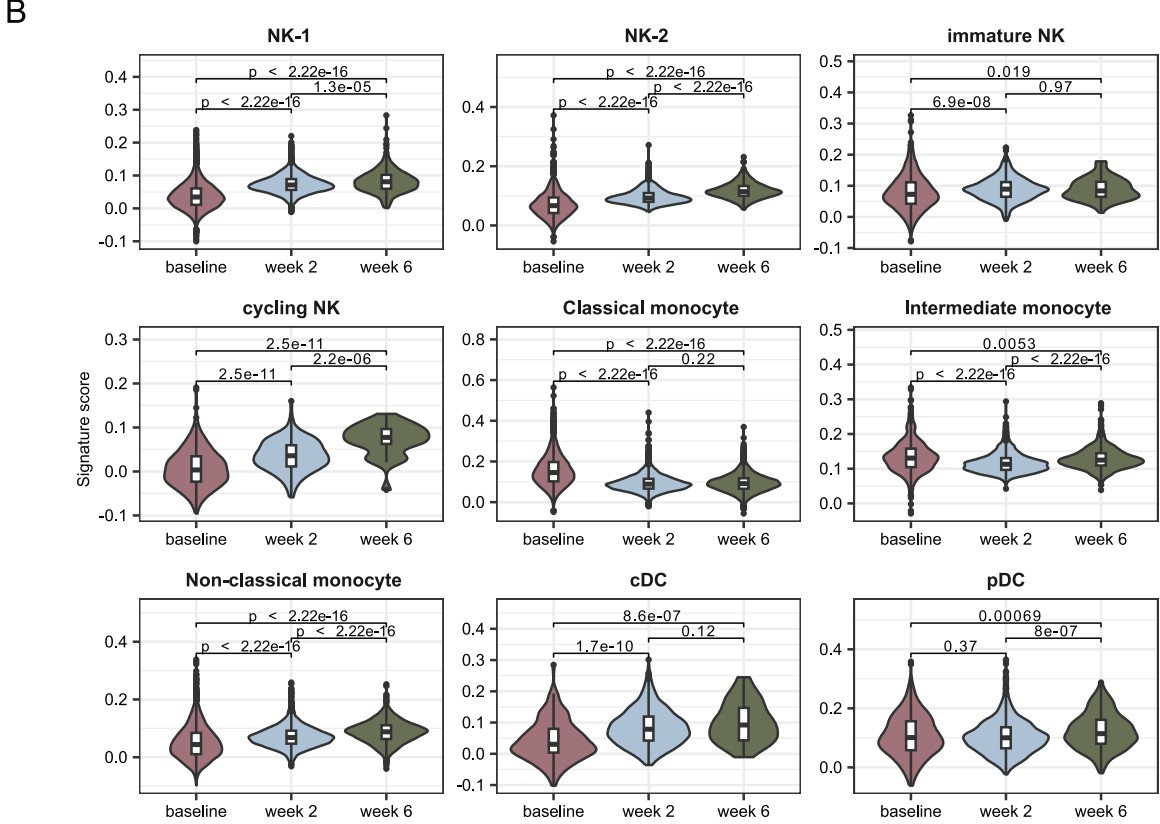

## Discussion

While binding and/or neutralizing antibody titers are most often used as readouts in vaccine studies to gauge immunogenicity and establish correlates of protection, a full analysis must include all the inter-dependent and interacting arms of the immune system. Doing so can shed light on how subtle interactions impact the overall response and hence clinical outcome. In this study, we comprehensively characterized the peripheral immune response induced by the mRNA-1273 vaccine in rhesus macaques. We found that a substantial fraction of the S-specific B cells elicited by the vaccine had a highly activated phenotype consistent with recent exit from GCs and which increased in frequency after the second dose. Furthermore, a small subset of S-specific T cells exhibited very high expression of key signaling genes including *CCL3*, *CCL4L1*, *IL10*, *IL21*, and *IFNG*, responsible for activation

**Fig. 8 | Upregulation of inflammatory pathways after second dose. A** Gene set enrichment analysis using Hallmark pathways from MSigDB *(34)*. A Kolmogorov-Smrinov test with the Benjamini-Hochberg correction for multiple testing was used to identify pathways with significant changes. Only pathways with at least one significant result are shown. Left panel, week 2 compared to baseline. Middle panel, week 6 compared to baseline. Right panel, week 6 compared to week 2. Within each panel, each column represents one of the transcriptional clusters identified in Fig. 6a. GSEA of sorted innate cells compared between different time points. **B** Signature score indicating the expression levels of 50 genes identified as being predictive of the magnitude of antibody responses to vaccines *(21)*. Signature scores were calculated using the ModuleScore function in Seurat. Boxes show the interquartile range, with the median marked as heavy horizontal band. Whisker represents the highest (lowest) datapoint within 1.5 times the interquartile range of the 75th (25th) percentile. A total of 30,671 cells from the baseline, 23,073 cells from week 2, and 7307 cells from week 6 were analyzed. P values were obtained using a two-sided unpaired Wilcoxon test with Holm correction for multiple testing. Mono monocyte, NK natural killer, pDC plasmacytoid dendritic cell, cDC classical dendritic cell. Source data are provided as a Source Data file.

of both adaptive and innate cell types. mRNA-1273 also stimulated differentiation of NK cells, inducing upregulation of genes related to antibody function, such as complement (*CFD* and *C1QB*) and the Fc receptor gene (*FCGR3*). Finally, most innate immune cells significantly upregulated a restricted set of genes which have been shown to be predictive of antibody responses to a variety of vaccines, with even higher expression of this module after the second dose. Importantly, we have only sampled cells from blood, while key immune responses also take place in the lymph nodes, spleen, and mucosa. We thus have not included component of the vaccine response that are unique to those sites. Despite this limitation, we show that the mRNA-1273 vaccine activates multiple components of the immune system both directly by antigen recognition, and indirectly by the production of cytokines and chemokines that may contribute to orchestrate the development of antibodies. Thus, although neutralizing antibodies represent a correlate of protection, the integrated action of multiple components of the immune system is required to achieve this outcome.

We identified by phenotype three subsets of S-specific memory B cells elicited by mRNA-1273 vaccination. A previous study found that SARS-CoV-2-specific class-switched resting memory B cells detected after vaccination were clonally related to IgM memory B cells identified from the time point of vaccination[66]. Notably, however, the levels of SHM were similar between the two groups, suggesting that the later RM cells were the result of extrafollicular expansion of the unswitched ancestors. We found a lack of increase in SHM consistent with this ontogeny and observed a substantially reduced frequency of RM cells after the second dose compared to the first.

Concomitant with the decline in RM was the expansion of B cells with a highly activated LZ-like phenotype. After the second dose, LZ-like B cells upregulated a number of activation pathways and expressed higher levels of TNF, suggesting an important signaling role in the extended innate immune activation we observed at the same time point. LZ-like B cells were clonally expanded and enriched for RBD-binding, although they did not have higher levels of somatic hypermutation at these early time points. These data suggest that the second dose of vaccine may recruit a fresh round of naïve B cells into germinal centers, likely with higher germline affinities for epitopes within RBD that are more divergent from previously encountered antigens[66,67]. As RBD is the dominant site for neutralization, the apparent de novo elicitation of these new clonal lineages by the second dose of vaccine may help explain why a third dose is required for acquiring breadth of neutralization against SARS-CoV-2 variants of concern[68,69].

As expected, S-specific T cells were also more frequent after the second dose. When stimulated, a subset of these cells showed increased expression of genes encoding cytokines and chemokines important for stimulation of innate immune responses, and the pathways regulated by those factors were indeed observed to be activated in innate cell types. Although we were unable to assess the phenotypes of vaccine-elicited T cells in detail, the expression of IL-21 is consistent with the bias toward Th1 or circulating Tfh populations observed in previous studies[2,13]. S-specific Tfh cells elicited by the first dose of vaccine likely play a key role in the recruitment of the new naïve B cells that gave rise to the LZ-like cells observed in response to the second

dose of vaccine. However, it remains an open question if T cells exhibit phenotypic shifts in response to the first versus the second dose, as we observed for B cells.

In the innate response, we found a shift of NK cells from NK-1 to the more mature NK-2 phenotype after the first dose of vaccine, which persisted after the second dose. Transient changes in peripheral NK cell populations have been reported after BNT162b2 vaccination[17,18], including a possible association with eventual neutralizing antibody titers[70], although not every study has reported these effects[20]. Differences in NK cell responses to a second vaccine dose may be driven by the stimulation of Fc receptors such as CD16 by immune complexes formed with antibodies produced in response to the previous exposure. However, there may also be a role for trained immunity[71]. In particular, activation of NK cells with IL-12 has been shown to result in enhanced IFNγ production upon restimulation[72], and we indeed observed a modest increase in the expression of *IFNG* (Fig. S8b) and upregulation of the Interferon Gamma Response pathway (Fig. 8a) in NK-2 cells at week 6 compared to week 2. Notably, IL-12 is also a key signaling molecule for driving the differentiation of naïve CD4 T cells toward a Th1 phenotype[73], which has also been associated with this vaccine[2,13]. Furthermore, the expression of an NK cell regulatory gene was associated with the production of broadly neutralizing antibodies against HIV-1[74], and NK cells have also been shown to boost antibody production in B cells after vaccination in mouse studies[75].

Trained immunity has also been reported in myeloid cells, despite their relatively short lifespan. In cynomolgus macaques vaccinated with MVA, a population of pro-inflammatory cells expressing high levels of CD11c, CD14, CD16, CD45, and IP-10 were expanded more after the second dose than the first[52]. In this study, we observed a similar trend toward the expansion of CD14⁺CD16⁺ intermediate monocytes at week 6 compared to week 2 (Fig. 7a) and increased expression of *ITGAX* (CD11c) (Fig. S8c). Notably, this subset also activated key signaling pathways at this time point, including IL6-Jak-Stat3, the Interferon Alpha Response, and the Interferon Gamma Response (Fig. 8a). Although neutrophils are not present in the PBMC samples used for this study, changes in the inflammatory profiles of those cells after booster vaccination have also been reported[52]. Combined with the phenotypic shifts we found in NK cell subsets, our work indicates that a two-dose primary regimen of mRNA-1273 synergistically activates multiple components of the innate immune response, likely with cooperative effects that also impact the quality of the adaptive immune response.

Several recent studies have used systems vaccinology approaches to analyze large multimodal datasets and produce insights into immune function[76–78]. These have identified different gene sets associated with different vaccine outcomes; however, reproducibility has generally been low across various vaccines and study conditions[21]. To mitigate this effect, a recent study considered 13 vaccines together, identifying a common gene set associated with a pro-inflammatory signature that was predictive of higher antibody responses across all vaccines considered[21]. Interestingly, while the original study found that the expression of inflammatory pathways returned to baseline by 7 days after vaccination, we found the same set of genes to be significantly upregulated by mRNA-1273 vaccination even 2 weeks after

vaccination, and this effect was even stronger for the second dose than for the first. Whether this is a property of the novel mRNA vaccination modality or a species difference between humans and macaques will need to be determined by future work. Additionally, due to the small and homogenous cohort examined here, it remains unclear whether the increase in the inflammatory signature at these later time points retains the correlation with vaccine outcomes observed in the original study. Notably, the pro-inflammatory signature was particularly associated with classical monocytes and cDCs[21], the former of which was one of only two cell types for which we observed a decreased expression of the pro-inflammatory module (Fig. 8b).

The possibility that immune activation elicited by the first vaccine dose may create a pro-inflammatory environment with beneficial effects on the antibody response induced by the second dose would have important ramifications for the design of clinical vaccine schedules. Trained immunity has been suggested to last at least several months[79]; however, it remains unknown how long the additive effects seen here in response to a second exposure might persist. Our data suggest at least that the four-week interval used for the primary series of mRNA-1273 vaccination falls within this window.

In summary, our data reveal a comprehensive picture of the immune responses elicited by the mRNA-1273 vaccine in NHP. Obviously, further studies in humans are necessary to confirm and to explore the clinical relevance of these findings. Our results show that a persistent pro-inflammatory state established by the first dose of vaccine likely gives rise to a qualitatively improved response to the second dose by activating signaling pathways that link innate and adaptive immunity in both directions. Furthermore, we found that vaccine-responsive T cells expressed cytokines important for stimulating multiple innate and adaptive cell types. This expression profile was also consistent with the polarization of these T cells toward a Th1 or Tfh phenotype, capable of enhancing germinal center function and the quality of the B cell response. Finally, we demonstrated the efficacy of the second dose of vaccine in producing highly activated B cells that target key epitopes for neutralization, which may serve as the substrate for broader reactivity in response to a booster vaccination. Overall, this study provides mechanistic insight into the dynamics of interactions between innate and adaptive immunity and how these connections are likely critical for the clinical benefit that the mRNA-1273 vaccine has imparted.

## Limitations of the study

A major limitation of this study is that we only sampled immune cells from peripheral blood, while critical immune responses take place in tissue sites, including the spleen, lymph nodes and mucosa. This study is also constrained by the lack of characterization of genetic variations which may have an impact on immune responses. Poor recovery of total mRNA including TCR transcripts reduced our ability to characterize the T cell response in these animals. The innate cell data required imputation to correct for sparseness, and the constraints of our flow cytometry panel (Table S2) meant that we could not characterize the absolute frequency of several cell types. Furthermore, with only two-time points, our study necessarily omits both the rapid kinetics of the initial innate immune response and the long-term affinity maturation of B cells. Finally, as the rhesus macaque genome is not fully annotated, with 23% of the genes in the reference transcriptome used in this study having unknown function, our ability to derive biological insight from the data was incomplete.

## Methods
### Study details
Experiments in animals were performed in compliance with National Institutes of Health (NIH) regulations and with approval from the Animal Care and Use Committee of the Vaccine Research Center and from Bioqual (Rockville, MD). Female and male Indian-origin rhesus

macaques were vaccinated intramuscularly at week 0 and at week 4 with 100 µg of mRNA-1273 in 1 ml of 1 Å- phosphate-buffered saline (PBS) into the right hind leg. Cryopreserved PBMCs from the 857.1 preclinical trial[7] were obtained at 1 week prior to vaccination ("baseline") and 2 weeks after both primary vaccinations (week 2 and week 6). For this study, we limited our investigation to the eight animals which received 100 µg doses of mRNA-1273 to match what was approved for human vaccination.

### Innate and B cell sort
Frozen rhesus macaque PBMCs were thawed into warm R10 media (RPMI + 10% Fetal Bovine Serum + 2 mL L-Glutamine, 100 U/ml penicillin and 100 µg/ml streptomycin; all reagents from Gibco) containing DNase I (MilliporeSigma), followed by one wash with R10 and one wash with FACS buffer (PBS with 2% FBS). For B and innate cell staining, cells were resuspended in 100 µL of Live/Dead Fixable Blue Dead Cell Stain Kit (Invitrogen, cat# L23105) diluted 1:200 in PBS for 10 min at room temperature. Cells were washed with FACS buffer and incubated for 20 min with the staining cocktail consisting of antibodies and probes. The antibodies used in the staining cocktail were: CD8-BV510 (Biolegend, clone RPA-T8, cat# 301048), CD56-BV510 (Biolegend, clone HCD56, cat# 318340), CD14-BV510 (Biolegend, clone M5E2, cat# 301842), CD16-BUV496 (BD Biosciences, clone 3G8, cat# 612944), CD3-APC-Cy7 (BD Biosciences, clone SP34-2, cat# 557757), CD19-PECy7 (Beckmann Coulter, clone J3-119, cat# IM36284), CD20 (BD Biosciences, clone 2H7, cat# 564917), CD27-BV605 (Biolegend, clone O323, cat# 302830), IgD-FITC (Southern Biotech, cat# 2030-02), IgG-Alexa Fluor 700 (BD Biosciences, clone G18-145, cat# 561296), IgM-PECF594 (BD Biosciences, clone G20-127, cat# 562539), CD11c-BUV395 (BD Biosciences, clone S-HCL-3, cat# 744440), CD123-BV786 (BD Biosciences, clone 7G3, cat #564196), HLA-DR-PECy5.5 (Invitrogen, clone TU36, cat# MHLDR18). For dilutions of antibodies see Table S2. Probes consisted of the SARS-CoV-2 proteins S-2P, S1, RBD, and NTD conjugated to APC, BV570, BV421 and BV711, respectively. Proteins were synthesized as described previously[80,81]. Additionally, each sample was labeled with 1 µl of TotalSeq-C Hashtag antibodies (Biolegend) that was incubated together with the staining cocktail. After incubation, cells were washed twice with FACS buffer and resuspended in R10 for sort.

From each sample, 10,000–60,000 innate cells were sorted into a tube containing FBS according to the gating strategy shown in Fig. S1B. From baseline samples, 50,000 naïve B cells were also sorted in a separate tube. From week 2 and 6 samples, antigen-specific cells were single-cell sorted into 96-well plates containing 5 µL of TCL buffer (Qiagen) with 1% β-mercaptoethanol for sequencing by SmartSeq. All sorts were performed using a BD FACSymphony S6 Cell Sorter (BD Biosciences) with BD FACSDiva Software version 9.5.1 (BD Biosciences) and data were analyzed using Flowjo v10.8.1.

### T cell stimulation and sort
Frozen rhesus macaque PBMC were thawed into warm R10 media (RPMI + 10% Fetal Bovine Serum + 2 mL L-Glutamine, 100 U/ml penicillin and 100 µg/ml streptomycin; all reagents from Gibco) containing DNase I (MilliporeSigma) and rested for 1 h at 37 °C/5% CO2. Cells were stimulated with two peptide pools corresponding to S1 and S2 of the vaccine insert SARS-CoV-2 S protein (JPT Peptide Technologies) at 2 µg/mL of each peptide for 6 h at 37 °C/5% CO2. A DMSO only control was included for each sample. Anti-CD154 BV421(Biolegend, clone 24-31, cat# 310824) was included during the 6-h culture and GolgiStop (BD Biosciences) was added after 2 h of stimulation. Following stimulation, cells were washed and stained with Aqua LIVE/DEAD dye (Thermo-Fisher) for 10 min, and subsequently stained with an antibody mix that included anti-CD69 FITC (Biolegend, clone FN50, cat# 310904), anti-CD28 ECD (Beckman Coulter, clone CD28.2, cat# 6607111), anti-CD4 PE-Cy7 (BD Biosciences, clone L200, cat# 560644), anti-CD3 APC (BD Biosciences, clone SP34-2, cat# 557597), anti-CD14 BV510 (Biolegend,

clone M5E2, cat# 301842), anti-CD20 BV510 (BD Biosciences, clone 2H7, cat# 563067), anti-CD95 BV650 (Biolegend, clone DX2, cat# 305642) and anti-CD8 BV785 (Biolegend, clone RPA-T8, cat# 301046). For dilutions of antibodies, see Table S3. In addition, cells were incubated with TotalSeq™ Hashtag antibodies (Biolegend) with separate antibodies applied to the DMSO-only condition and the stimulated conditions. For each animal, 1000-2000 memory CD4 cells from the DMSO only condition were sorted and combined with memory cells expressing CD154 and CD69 from each S pool stimulation for processing for 10× Genomics. All sorts were performed using a BD FACSymphony S6 cell sorter and data were analyzed using Flowjo v10.8.1.

### Single-cell sequencing by 10x genomics
Bulk-sorted cells were pooled into a tube and loaded on the 10x Genomics Chromium Chip according to the manufacturer's protocol for the Next GEM Single Cell 5' Kit v1.1 (10x Genomics, PN-1000165). To sequence single-cell gene expression and cell surface oligonucleotides (from Hashtag antibodies), libraries were prepared according to manufacturer's instructions using the Chromium Single Cell 5' Library Construction Kit (10x Genomics, PN-1000020) and Chromium Single Cell 5' Feature Barcode Library Kit (10x Genomics, PN-1000080), respectively. To sequence TCR repertoire libraries were prepared using the Chromium Single Cell V(D)J Enrichment Kit (10x Genomics, PN- PN-1000005). These libraries were sequenced on an Illumina NovaSeq platform (Illumina). To sequence BCR repertoire, heavy and light chains were amplified from the cDNA using IgG_REV, IgA_REV, IgK_REV, or IgL_REV primers (Table S4) with the addition of Illumina sequences as detailed in[25]. These Illumina-ready libraries were sequenced using 2 × 300 paired-end reads on an Illumina MiSeq.

### Processing of transcriptomic data from 10x chromium
The counts matrix was generated with Cellranger v3.1.0, using the protein-coding genes of the rhesus macaque reference genome from Ensembl (Mmul10, release #101). The filtered feature/barcode matrices were imported into Seurat v4.0.1 using R v4.0.4. Cells absent from either the gene expression or antibody capture datasets were eliminated, along with cells expressing zero counts of hash oligos. For hash demultiplexing, Seurat's `HTOdemux` function was applied, and only singlets were retained for subsequent analyses. The filters based on percent mitochondrial genes and number of genes expressed varied between sample, but the upper threshold for percent mitochondrial genes ranged from 4 to 10%; the minimum for number of genes expressed was 200 genes for each sample; the upper threshold for number of genes expressed ranged from 1500 to 2500. The raw counts of each individual sample were merged.

For the innate cell sorts, clusters of contaminating B and T cells (identified using Seurat's `FindClusters` and `RunUMAP`) were removed and not included in downstream analysis. The merged and filtered data matrix was then corrected for sparseness using DeepImpute[54] with tensorflow v2.1.0 and default parameters. The imputed matrix was then re-imported into Seurat.

For both innate and T cells, Seurat was used to normalize the (imputed) count matrix, find variable genes, and regress out technical factors. Principle compents (PCs) were then calculated and used as input to Harmony[45] to remove batch effects between samples. The integrated output was clustered using `FindClusters` and marker genes were identified with `FindAllMarkers`. Canonical marker genes were used to annotate the detected clusters.

Gene set enrichment analysis of the innate cell data was conducted using the R library fgsea[82]. The data were broken down by cell type and time point, and differential expression analysis of all 21,369 genes was conducted using FindMarkers between each subset. Finally, the statistically significant (adjusted $p < 0.05$) pathways were detected by inputting the genes, ranked by average log fold change, into the function `fgsea`. The Hallmark gene set databases originated from MSigDB[35]. GSEA results were displayed using code adapted from[61].

Signature scores for 50 pro-inflammatory genes (Table S1) previously determined to be predictive of antibody responses to a variety of vaccines[21] were calculated using `AddModuleScore` in Seurat.

### T cell V(D)J analysis
cellranger vdj (10x Genomics) was used to assign TCR annotations for each read-pair. TCR alpha recovery was minimal and not followed up further. An in-house shell script was used to filter for productive TCR betas. All subsequent analyses were done in R (v4.0.2). Cells with duplicated barcodes were discarded, and the TCR data was merged with the T cell transcriptomics Seurat (v4.0.1) object metadata for hash demultiplexing. Usage of each TRBV gene was compared between the S-specific and non-specific T cells from each animal using a paired samples Wilcoxon test in R.

### Single-cell sequencing by SmartSeq
Full-length transcriptomes were sequenced using a modified SMART-Seq protocol as in ref. 83. Heavy and light chain variable regions were enriched by amplifying cDNA with a forward primer complementary to the TSO and IgA/IgG_REV or IgK/IgL_REV primer pools (Table S3) as in ref. 25.

### Preprocessing of transcriptomic data from SmartSeq
Sequencing reads were trimmed with Trimmomatic v 0.39[84], trimming nucleotides with Phred score below 20 and removing reads with resulting length less than 50 nucleotides. Alignment of reads to macaque protein-coding reference build Mmul10 was performed using Spliced Transcript to a Reference (STAR) v. 2.5.1b[85]. Gene quantifications of reads as transcripts per million were computed with Stringtie v. 1.33. Individual cells were removed from the analysis based on trimming and alignment metrics, as well as read quantity, feature quantity, quantity of mitochondrial reads, and diversity of transcripts.

Seurat was used to calculate principal component analysis, which was used as input to integrate the data by batch with Harmony[45]. PCA, cell clustering and UMAP dimensionality reductions were conducted in Seurat. A single cluster was identified as likely T cells by its more than triple the expression of *CD3E* than the next highest cluster and so was removed from the analysis. Clustering and dimensionality reduction were repeated after this cell filter.

Differential expression of cell clusters was conducted with Seurat's function `FindAllMarkers`, and `FindMarkers` was used for comparison of time points week 2 and week 6. Those results were used as input to Gene set enrichment analysis conducted with the R package fgsea[82] and the hallmark pathways from MSigDB[35].

### B cell V(D)J analysis
Full-length V(D)J sequences were reconstructed using BALDR[86] and filtered using `filterBALDR.pl` (https://github.com/scharch/filterBALDR). Gene annotation and clonal assignment were done using SONAR v4.2[87] in single cell mode. Clonal flow diagrams were plotted using the `ggalluvial` package in R v4.2.1. Public clone were identified by clustering CDR H3 amino acid sequences using usearch[88] `cluster_fast` to cluster CDR H3 amino acid sequences at 80% identity. Members of the human public clone were identified from CoVAbDab[89] sequences downloaded as of January 24th, 2023 meeting the following criteria: (1) derived from *IGHV3-30*, *IGHV3-30-3*, or *IGHV3-30-5*; (2) 14 amino acid CDR H3 with Gly at position 6 and Tyr at position 8; (3) originating from a human; (4) *not* annotated as binding to any RBD or NTD epitope. This resulted in 326 public clone sequences, which were plotted using ggseqlogo[90].

## Reporting summary

Further information on research design is available in the Nature Portfolio Reporting Summary linked to this article.

## Data availability

Hallmark pathways for gene-set enrichment analysis were downloaded from https://www.gsea-msigdb.org/gsea/msigdb/human/genesets.jsp?collection=H. All raw and processed sequencing data generated in this study have been deposited in the Gene Expression Omnibus under series GSE232117 at https://www.ncbi.nlm.nih.gov/geo/query/acc.cgi?acc=GSE232117. Source data are provided with this paper.

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

## Acknowledgements

The authors thank Amy Noe, Dillon Flebbe, Nadesh Nji, Evan Lamb, and other members of the VRC NHP Immunogenicity Core for processing and providing samples for this study. The authors also thank Arun Boddapati, Amit Upadhyay, Gregory Tharp, and Steven Bosinger for help discussions. The authors also thank Tracy Ruckwardt and Emily Phung for the peptide pools used to stimulate T cells and conjugating probes used to sort antigen-specific B cells. This work was funded in part by the Intramural Research Program of the Vaccine Research Center, National Institute of Allergy and Infection Disease, National Institutes of Health.

## Author contributions

Conceptualization: C.A.S., N.S.L., R.A.S., D.C.D. Data curation: C.A.S., L.P. Formal Analysis: C.A.S., D.M., L.P., N.S.L., C.W., K.L.B., S.W.D. Investigation: D.M., N.S.L., K.L.B,. A.R.H., F.L., D.A. Resources: I.T.T., K.E.F., A.C., D.K.E., R.A.S. Supervision: C.A.S., P.D.K., R.A.K., R.A.S., D.C.D. Visualization: C.A.S., L.P., N.S.L., C.W., K.L.B. Writing – original draft: C.A.S., L.P., N.S.L., C.W., D.C.D. Writing—review & editing: all authors.

## Funding

## Competing interests

A.C. and D.K.E. are employees of and shareholders in Moderna, Inc. The remaining authors declare no competing interests.
