## [Peer Review File · Nature Communications]

Interaction Dynamics Between Innate and Adaptive Immune Cells Responding to SARS-CoV-2 Vaccination in Non-Human PrimatesREVIEWER COMMENTS

Reviewer #1 (Remarks to the Author):

This study by Schramm et al. describes the transcriptional landscape of immune cell subsets from rhesus macaques vaccinated with mRNA-1273 (Moderna). The strength of the work is in the comprehensive characterization of nearly all immune subsets in the PBMC of vaccinated macaques. Limitations are that only PBMC are interrogated and time points are limited. My overall biggest concern is that the authors may be over-interpreting some of their data.

Major comments:

1. The results section should be trimmed to remove speculation about cell-cell interactions and hypotheses. No functional analyses of cell-cell interactions were performed and only two time points total are assessed. For instance, "These results demonstrate the effects of vaccine-induced CD4 T cells to promote B cell differentiation into antibody-producing plasma cells, GC development, and to induce activation of innate cells, suggesting a late cross-talk between the adaptive and innate immune responses." seems more conjecture than actually supported by the data.
2. Another major limitation of the study is the fact that only PBMC is assessed, while lymph nodes and spleen (where GC are present) is not. This should be directly mentioned in the discussion and also added to the section "Limitations of the study".
3. Authors reference paper (ref 21) where a transcriptomic comparison of 13 different vaccines was conducted. One of the major things I would have loved to see what some sort of comparator group receiving recombinant protein or viral vectored vaccine so we can start to say how the mRNA vaccines are different. Can the authors at least build out this discussion further and indicate how comparable your data is to the data collected in Ref 21. Will readers be able to make any comparisons between the studies? Were similar methods used?
4. How common is the following definition? "We next identified public clones, defined here as IgH sequences from different animals using the same VH gene and having at least 80% amino acid identity in CDR H3." Is there a BCR functional basis for this definition?

Minor comments:

1. Fig 2b and 2c are swapped in the manuscript.
2. Two of the rows are identical in the table in Fig 2e. Unsure what this represents.
3. Fig. 2e, f: It is unclear how prevalent these public clonotypes are based on how the data are presented. Are these each found in a single B cell? Or are they more immunodominant following clonal expansion?
4. Fig. 3b: x-axis is offset from ticks
5. Fig. 3e and 3f are swapped in the manuscript.
6. Fig. 7b is a panel in the manuscript, but is never mentioned in the text.

Reviewer #2 (Remarks to the Author):

This manuscript provides a comprehensive analysis of adaptive, humoral and innate immune responses in rhesus macaques receiving two separate doses of the Moderna mRNA 1273 vaccine against SARS-CoV-2. These studies demonstrate the cross species similarity of responses in rhesus macaques and those found in humans. Antibody titers were found to increase between 3 and 6 months with concomitant increase in hypermutation of antibody genes. T cell responses were also stable through 6 months and highly cross reactive with later variants including Omicron. Second doses 1 month later induced increases in circulating germinal center-like B cells. IFN gamma and IL-2 STAT signaling as well as TNF signalling were also markedly increased by the second immunization, contributing to the anamnestic response. B cells subsets were shown to have recently emigrated from germinal centers by transcriptomic analysis. Transcriptomic analysis revealed a decrease in classical monocytes with a moderate increase in intermediate monocytes after immunization. These results are consistent with previous observations with the BNT162b2 vaccine. This population is thought to be highly active in antigen presentation and cytokine production, and expresses increased CCR5 levels. These studies further characterized NK cell clusters, with a shift toward the more mature NK-2 phenotyped as well as pDC and cDC. This is an important study providing insights regarding the interaction between innate and adaptive immunity, particularly with the role of intermediate monocytes in trained immunity.

Reviewer #3 (Remarks to the Author):

Schramm et al. have analysed the adaptive and innate immune responses induced by a single dose or a 2-dose regimen of mRNA-1273 vaccination in rhesus macaques. The data are complementary to a previous publication of vaccine-induced immune readouts assessed in the same animals, which is extended by the present study with an in-depth immunological and transcriptome assessment. Characterization of B cell reactivities in the present study showed parallels with published human data after vaccination with mRNA-1273 and the study defines distinct activated B cell subsets that exit germinal centers after vaccination. Transcriptome analysis of antigen-specific CD4 T cell revealed four CD4+ subsets that are differentiated by expression of genes known to modify innate immune responses. A transcriptomic analysis of innate immune cells at baseline and 2 weeks after each vaccination revealed 10 innate immune cell clusters and the dynamic changes in the frequency of these subsets during the study were analyzed. Finally, several innate immune cell subsets showed differential regulation of a vaccine-predictive inflammatory gene set that was identified in an independent human study.

Strengths:

The manuscript is well written and clearly structured. The study is of general immunology and vaccinology interest but may be of limited relevance to the current COVID-19 vaccine field since it has been conducted in naïve NHP, while the majority of the human populations has been COVID-19 vaccinated, infected, or both.

Potential weaknesses / limitations:

The study is primarily descriptive and does not experimentally proof cross-talk between adaptive and innate immune responses, which could be achieved for example by CD4 T cell depletion in a subset of animals before second immunization.

The study is restricted to PBMC samples although the advantage of preclinical over clinical studies is retrievability of other sample types (e.g. mucosal samples such as BAL, LN biopsies).

While the antigen-specific IgH chain repertoire is analyzed, this was not performed for the light chain repertoire.

Specific points:

1. Please include N and stats methods in all figure legends.
2. Please describe in more detail what is shown in Figure 2a eg by using a more specific y axis label.
3. Figure labels 2b and 2c are switched between manuscript and figures. Please include detail about how data are displayed and statistically analysed. Since data are not unrelated within one animal, a representation of median CDR H3 length and % SHM per animal seems appropriate and should be provided. This should also be taken into account for the statistical analysis
4. Figure labels 3e and 3f are switched between manuscript and figures.
5. Line 108: 'The closest human homolog of this gene is IGHV3-30 (Fig S3)'; According to figure S3, IGHV3-33 is the closest human homolog, please clarify/correct
6. It would be useful for the reader to provide % identity/homology to human clone in table F of Fig. 2.
7. Lines 135-138: 'In agreement with the identification of LZ-like cells as having recently exited from germinal centers (GCs), this cluster was enriched for RBD- and NTD-binding B cells (Fig 3d), from which most neutralizing antibodies are derived, unlike those targeting epitopes in S2 (32).' The sentence suggests this result is expected based on recent exit from GCs, but it is unclear why/how an epitope bias is resulting from GC exit; please provide a more detailed explanation.
8. Lines 160-162: The text suggests that a causal relationship between vaccine-induced cytokine responses and cooperation between humoral immunity and other immune cell types exists. While this relationship seems a likely explanation, this should not be positioned as 'proven' since other mechanisms (epigenetics, trained immunity) may also play a role here. I would be useful for the reader to provide and discuss potential alternative explanations for the observations.
9. Lines 221-224: Data are interpreted as demonstrating a direct effect of vaccine induced CD4 T cells on B cell differentiation. In my understanding there is no experimental proof of this direct effect. As in the previous point, the data suggest this, taking also published literature into account, but there is no experimental evidence, and this should be reflected in the interpretation of the data in the manuscript.
10. Lines 324-326: For clarity, perhaps reference Figure 7b here.
11. Lines 326-228: In my understanding the data provided here are rather contradicting the findings from the original study. Based on the study by Fourati et al., the classical monocyte scores should be highest after the second immunization, since the first dose would stimulate inflammatory responses which predispose the host to even higher immune responses after the second vaccine dose. Please clarify in the manuscript text.
12. Lines 329-330: There is a reasonable difference in immune responses between the first and second dose (Fig. 1b). If innate immune responses are influencing adaptive immunity as suggested here, the innate immune endotype (according to Fourati et al.) observed at week 2 should predict the increase in antibody responses observed at week 6. It would be interesting for the reader to include a correlation analysis of these readouts.

REVIEWER COMMENTS

Reviewer #1 (Remarks to the Author):

This study by Schramm et al. describes the transcriptional landscape of immune cell subsets from rhesus macaques vaccinated with mRNA-1273 (Moderna). The strength of the work is in the comprehensive characterization of nearly all immune subsets in the PBMC of vaccinated macaques. Limitations are that only PBMC are interrogated and time points are limited. My overall biggest concern is that the authors may be over-interpreting some of their data.

We thank the reviewer for taking the time to review our manuscript and helping us to strengthen it. We have edited the manuscript to avoid over-interpretation, as detailed in response to the reviewer's specific comments below.

Major comments:

1. The results section should be trimmed to remove speculation about cell-cell interactions and hypotheses. No functional analyses of cell-cell interactions were performed and only two time points total are assessed. For instance, "These results demonstrate the effects of vaccine-induced CD4 T cells to promote B cell differentiation into antibody-producing plasma cells, GC development, and to induce activation of innate cells, suggesting a late cross-talk between the adaptive and innate immune responses." seems more conjecture than actually supported by the data.

We agree that some of our speculations were overly exuberant. The particular suggestion of effects on CD4 T-B cell interactions has been stricken and we have also removed conjecture about antigen presentation by innate immune cells that had been in line 309 (as edited).

2. Another major limitation of the study is the fact that only PBMC is assessed, while lymph nodes and spleen (where GC are present) is not. This should be directly mentioned in the discussion and also added to the section "Limitations of the study".

We agree and have added text to this effect in lines 356-358 and 460-461.

3. Authors reference paper (ref 21) where a transcriptomic comparison of 13 different vaccines was conducted. One of the major things I would have loved to see what some sort of comparator group receiving recombinant protein or viral vectored vaccine so we can start to say how the mRNA vaccines are different. Can the authors at least build out this discussion further and indicate how comparable your data is to the data collected in Ref 21. Will readers be able to make any comparisons between the studies? Were similar methods used?

We thank the reviewer for this suggestion and have added a comparison of the two studies in lines 426-436 of the Discussion.

4. How common is the following definition? "We next identified public clones, defined here as IgH sequences from different animals using the same VH gene and having at least 80% amino acid identity in CDR H3." Is there a BCR functional basis for this definition?

Unlike for TCRs, there is unfortunately no consensus definition for a B cell public clone, but the one we use is a relatively common and accepted one. We have added details of why we chose this threshold to

lines 106-109.

Minor comments:

1. Fig 2b and 2c are swapped in the manuscript.

Thanks, we have corrected this.

2. Two of the rows are identical in the table in Fig 2e. Unsure what this represents.

They represent two cells with the same CDRH3. We have added clarification of this to the figure legend.

3. Fig. 2e, f: It is unclear how prevalent these public clonotypes are based on how the data are presented. Are these each found in a single B cell? Or are they more immunodominant following clonal expansion?

We have added text on lines 110-113 to better describe why these public clones are of interest. While the reviewer correctly notes that they are not immunodominant in our data, they nonetheless represent a convergence that is likely to signify a real functional enrichment. We have also added a comparison to the naïve data from these animals to help make this point.

4. Fig. 3b: x-axis is offset from ticks

Thanks, we have fixed this.

5. Fig. 3e and 3f are swapped in the manuscript.

Thanks, we have fixed this.

6. Fig. 7b is a panel in the manuscript, but is never mentioned in the text.

Thanks, a call-out has been added to line 330

Reviewer #2 (Remarks to the Author):

This manuscript provides a comprehensive analysis of adaptive, humoral and innate immune responses in rhesus macaques receiving two separate doses of the Moderna mRNA 1273 vaccine against SARS-CoV-2. These studies demonstrate the cross species similarity of responses in rhesus macaques and those found in humans. Antibody titers were found to increase between 3 and 6 months with concomitant increase in hypermutation of antibody genes. T cell responses were also stable through 6 months and highly cross reactive with later variants including Omicron. Second doses 1 month later induced increases in circulating germinal center-like B cells. IFN gamma and IL-2 STAT signaling as well as TNF signalling were also markedly increased by the second immunization, contributing to the anamnestic response. B cells subsets were shown to have recently emigrated from germinal centers by transcriptomic analysis. Transcriptomic analysis revealed a decrease in classical monocytes with a moderate increase in intermediate monocytes after immunization. These results are consistent with previous observations with the BNT162b2 vaccine. This population is thought to be highly active in antigen presentation and cytokine production, and expresses increased CCR5 levels. These studies further characterized NK cell clusters, with a shift toward the more mature NK-2 phenotype as well as

pDC and cDC. This is an important study providing insights regarding the interaction between innate and adaptive immunity, particularly with the role of intermediate monocytes in trained immunity.

We thank the reviewer for the kind remarks in support of our manuscript.

Reviewer #3 (Remarks to the Author):

Schramm et al. have analysed the adaptive and innate immune responses induced by a single dose or a 2-dose regimen of mRNA-1273 vaccination in rhesus macaques. The data are complementary to a previous publication of vaccine-induced immune readouts assessed in the same animals, which is extended by the present study with an in-depth immunological and transcriptome assessment. Characterization of B cell reactivities in the present study showed parallels with published human data after vaccination with mRNA-1273 and the study defines distinct activated B cell subsets that exit germinal centers after vaccination. Transcriptome analysis of antigen-specific CD4 T cell revealed four CD4+ subsets that are differentiated by expression of genes known to modify innate immune responses. A transcriptomic analysis of innate immune cells at baseline and 2 weeks after each vaccination revealed 10 innate immune cell clusters and the dynamic changes in the frequency of these subsets during the study were analyzed. Finally, several innate immune cell subsets showed differential regulation of a vaccine-predictive inflammatory gene set that was identified in an independent human study.

Strengths:

The manuscript is well written and clearly structured. The study is of general immunology and vaccinology interest but may be of limited relevance to the current COVID-19 vaccine field since it has been conducted in naïve NHP, while the majority of the human populations has been COVID-19 vaccinated, infected, or both.

We thank the reviewer for the generous description of our manuscript.

Potential weaknesses / limitations:

The study is primarily descriptive and does not experimentally proof cross-talk between adaptive and innate immune responses, which could be achieved for example by CD4 T cell depletion in a subset of animals before second immunization.

We agree and have made changes to limit our speculation about cell-cell interactions as detailed in response to Reviewer 1 above. The particular suggestion of effects on CD4 T-B cell interactions has been stricken and we have also removed conjecture about antigen presentation by innate immune cells that had been in line 309 (as edited).

The study is restricted to PBMC samples although the advantage of preclinical over clinical studies is retrievability of other sample types (e.g. mucosal samples such as BAL, LN biopsies).

We agree and have noted this limitation as detailed in response to Reviewer 1 above and have added text to this effect in lines 356-358 and 460-461.

While the antigen-specific IgH chain repertoire is analyzed, this was not performed for the light chain repertoire.

We apologize for the oversight and have added parallel analysis of the light chain repertoire in Figs S4 and S5.

In addition, during the course of these revisions we discovered a minor error in the statistical calculations for IGH and TCRB V gene usage. Changes to Fig 2D have been made and do not affect the description in the text. The corrected version of Fig 4B now shows differential usage for 6 genes and the text on lines 190-192 has been updated to match.

Specific points:

1. Please include N and stats methods in all figure legends.

We have corrected these omissions and added the required details to all figure legends.

2. Please describe in more detail what is shown in Figure 2a eg by using a more specific y axis label.

We have changed the y-axis label to “cumulative number of cells” to reflect that the fact that each individual lineage only comprises a small number of cells. Perhaps more importantly, we have added a color legend to this panel and significantly increased the detail of the description in the accompanying caption.

3. Figure labels 2b and 2c are switched between manuscript and figures. Please include detail about how data are displayed and statistically analysed.

Thanks, the labels have been corrected and the statistical details have been added.

Since data are not unrelated within one animal, a representation of median CDR H3 length and % SHM per animal seems appropriate and should be provided. This should also be taken into account for the statistical analysis

Please allow us to explain why this approach may not be appropriate in this case: CDR H3 length is independently generated for each lineage within an animal during VDJ rearrangement of the pre-B cell ancestor. Because it is a property of each lineage, we use only one cell per lineage to create and compare these distributions. SHM is moderately correlated for cells in a single lineage, due to the germinal center history of common ancestors, but the selection of multiple daughter cells into the memory pool contains useful information. For SHM we thus include all cells. For completeness, we have added a new Fig S3 which shows the comparison for both statistics on a per animal basis, as the reviewer has requested. The trends are consistent across animals and the same as the aggregated data, though in some cases we lose the statistical power necessary to ascertain significance. We hope this is clear now.

4. Figure labels 3e and 3f are switched between manuscript and figures.

Thanks, this has been corrected.

5. Line 108: 'The closest human homolog of this gene is IGHV3-30 (Fig S3)'; According to figure S3, IGHV3-33 is the closest human homolog, please clarify/correct

We apologize for the imprecision; this has been clarified on lines 114-115

6. It would be useful for the reader to provide % identity/homology to human clone in table F of Fig. 2.

Unfortunately this is not possible as, by definition, a public clone is group of similar sequences, not a single fixed sequence. That is why we have represented the human public clone in Fig 2F using a sequence logo, which demonstrates that the conserved positions of the human public clone are likewise conserved in the rhesus public clone we observed. We have clarified this in the figure caption.

7. Lines 135-138: 'In agreement with the identification of LZ-like cells as having recently exited from germinal centers (GCs), this cluster was enriched for RBD- and NTD-binding B cells (Fig 3d), from which most neutralizing antibodies are derived, unlike those targeting epitopes in S2 (32).' The sentence suggests this result is expected based on recent exit from GCs, but it is unclear why/how an epitope bias is resulting from GC exit; please provide a more detailed explanation.

We apologize for the lack of clarity and have rewritten lines 140-145 to better convey our intent. Essentially: Neutralizing antibodies appear primarily after the second dose; the LZ-like B cells are also enriched after the second dose and focus on epitopes associated with neutralization; therefore LZ-like B cells may well have still been in GCs when the second dose was administered and hence would have only recently exited.

8. Lines 160-162: The text suggests that a causal relationship between vaccine-induced cytokine responses and cooperation between humoral immunity and other immune cell types exists. While this relationship seems a likely explanation, this should not be positioned as 'proven' since other mechanisms (epigenetics, trained immunity) may also play a role here. I would be useful for the reader to provide and discuss potential alternative explanations for the observations.

We agree and have noted these possibilities in lines 169-170 (as edited).

9. Lines 221-224: Data are interpreted as demonstrating a direct effect of vaccine induced CD4 T cells on B cell differentiation. In my understanding there is no experimental proof of this direct effect. As in the previous point, the data suggest this, taking also published literature into account, but there is no experimental evidence, and this should be reflected in the interpretation of the data in the manuscript.

As detailed above and in response to Reviewer 1, we agree, and have removed this hypothesis to avoid over-stating our findings.

10. Lines 324-326: For clarity, perhaps reference Figure 7b here.

Yes, thank you, we have added the Figure call-out as suggested.

11. Lines 326-228: In my understanding the data provided here are rather contradicting the findings from the original study. Based on the study by Fourati et al., the classical monocyte scores should be highest after the second immunization, since the first dose would stimulate inflammatory responses which pre-dispose the host to even higher immune responses after the second vaccine dose. Please clarify in the manuscript text.

The reviewer is correct, and we have reworded the sentence and the following ones to more accurately describe the results of Fourati et al. As noted in response to Reviewer 1, we have also have added a direct comparison of our results to those of Fourati et al in lines 426-436 of the Discussion.

12. Lines 329-330: There is a reasonable difference in immune responses between the first and second dose (Fig. 1b). If innate immune responses are influencing adaptive immunity as suggested here, the innate immune endotype (according to Fourati et al.) observed at week 2 should predict the increase in antibody responses observed at week 6. It would be interesting for the reader to include a correlation analysis of these readouts.

As noted in line 333-335 (as edited) the cohort is too small and homogeneous for us to detect any predictive power of this gene module. For the reviewer's benefit, we present Fig R1 below, showing the negative results.

Figure R1. Correlations between pro-inflammatory gene set module score and immune outcomes.

pre/wk2/wk6 F = Median Fourati gene module score across all innate immune cells in a particular animal at the indicated time point.

pre/wk2/wk6 F mono = Median Fourati gene module score across all classical monocytes in a particular animal at the indicated time point.

wk2/wk6 IgM/IgG cells = Frequency of antigen-specific cells of the indicated isotype in a particular animal at the indicated time point.

IgG cells ratio = Fold increase in the frequency of antigen-specific IgG⁺ B cells from week 2 to week 6 in a particular animal.

pctLZ wk2/wk6 = Percentage of B cell transcriptomes in the LZ-like cluster in a particular animal at the indicated time point.

Serological parameters measured at week 8 and reported in Appendix 2 of Corbett et al NEJM 2020:

IgG bind = Binding IgG AUC; nanoluc/pseudoneut = IC50 of neutralization by the indicated assay; ace2 = ACE2 binding inhibition; rbd/ntd bind = log10 AUC of binding to the indicated domain by MSD.

REVIEWER COMMENTS

Reviewer #3 (Remarks to the Author):

Thank you for the responses to comments and adjustments of the manuscript. Please find my additional comments below.

- Please adjust lines 446-448 ('...Our results show that a persistent pro-inflammatory state established by the first dose of vaccine yields a qualitatively improved response to the second dose,...') to reflect that a direct effect of pro-inflammatory state on second dose vaccine response has not been experimentally proven.
- Since TRBV3-4 (please mind the typo in revised text) was highly significantly enriched in S-specific cells, please include an assessment of TRBV3-4 -homologous gene usage enrichment in humans.
- Please include info whether one or all cells per lineage were included in the SHM and CDR H3 analysis diagrams (2b and 2c) in the figure legend. Furthermore, it is not clear why these two readouts should be handled differently if both are correlated within one lineage. It is of interest to the reader if a different analysis approach leads to a different outcome, therefore please include both analyses (one/all cells per lineage) for SHM and CDR H3 length in the manuscript / supplement.
- Lines 114-115: The cited literature shows only a clear enrichment for IGHV3-30 and IGHV3-30-3, please update/correct

Reviewer #3 (Remarks to the Author):

Thank you for the responses to comments and adjustments of the manuscript. Please find my additional comments below.

- Please adjust lines 446-448 ('...Our results show that a persistent pro-inflammatory state established by the first dose of vaccine yields a qualitatively improved response to the second dose,...') to reflect that a direct effect of pro-inflammatory state on second dose vaccine response has not been experimentally proven.

Done.

- Since TRBV3-4 (please mind the typo in revised text) was highly significantly enriched in S-specific cells, please include an assessment of TRBV3-4 -homologous gene usage enrichment in humans.

Thanks for catching the typo. We have also added the requested assessment.

- Please include info whether one or all cells per lineage were included in the SHM and CDR H3 analysis diagrams (2b and 2c) in the figure legend.

We apologize for the oversight and have added the necessary detail to the figure legend.

Furthermore, it is not clear why these two readouts should be handled differently if both are correlated within one lineage.

We apologize for not having explained the difference clearly enough. CDR H3 length is fixed at rearrangement (<https://doi.org/10.1371/journal.pone.0036750>) and for most practical purposes does not vary at all within a lineage (< 2% of cells have SHM-associated indels and they are concentrated in cells with high total SHM: <https://www.nature.com/articles/gene201228>). Thus, we treat it as a per-lineage variable and plot accordingly. In contrast, the level of correlation in SHM between two cells depends on the number of cell divisions since their most recent shared ancestor. In peripheral memory the correlation is typically fairly weak. In addition, when cells from the same lineage have different levels of SHM, the arbitrary selection of one of them to represent the lineage will result in different possible distributions and likely confound the statistical analysis.

It is of interest to the reader if a different analysis approach leads to a different outcome, therefore please include both analyses (one/all cells per lineage) for SHM and CDR H3 length in the manuscript / supplement.

We have added these panels to Figs. S3, S4, and S5.

- Lines 114-115: The cited literature shows only a clear enrichment for IGHV3-30 and IGHV3-30-3, please update/correct

We thank the reviewer for catching this. These 4 genes are highly homologous, to the point where antibodies derived from one can frequently be misassigned to another during repertoire analysis. It is very likely that they are functionally interchangeable, but the reviewer is correct to point out that this has not been formally demonstrated. We have thus corrected the noted line.